

# Boreal fire records in Northern Hemisphere ice cores: A review

Michel Legrand[1], Joseph McConnell[2], Hubertus Fischer[3], Eric W. Wolff[4], Susanne Preunkert[1], Nathan Chellman[2], Daiana Leuenberger[3], Olivia Maselli[2], Michael Sigl[2], Simon Schüpbach[3], and Mike Flannigan[5]

[1]Université Grenoble Alpes, CNRS, Laboratoire de Glaciologie et Géophysique de l'Environnement (LGGE), Grenoble, France.
[2]Desert Research Institute, Nevada, USA.
[3]Climate and Environmental Physics, Physics Institute & Oeschger Centre for Climate Change Research, University of Bern, CH.
[4]Department of Earth Sciences, University of Cambridge, UK.
[5]University of Alberta, Department of Renewable Resources, Edmonton, AB, Canada.

*Correspondance to*: Michel Legrand (legrand@lgge.obs.ujf-grenoble.fr)

**Abstract.** Here we review different attempts made since the early 1990s to reconstruct past forest fire activity using chemical signals recorded in ice cores extracted from the Greenland ice sheet and at a few mid-northern latitude, high-elevation glaciers. We first examined the quality of various inorganic (ammonium, nitrate, potassium) and organic (black carbon, various organic carbon compounds including levoglucosan and numerous carboxylic acids) species proposed as fire proxies in ice, particularly in Greenland. We discuss limitations in their use with respect to the considered time period (recent versus pre-industrial times), their atmospheric lifetime, and the relative importance of other non-biomass burning sources. Different high-resolution records obtained at several Greenland drill sites and covering various timescales, including the last century and Holocene, are discussed. We explore the extent to which atmospheric transport can modulate the record of boreal fires from Canada as recorded in Greenland ice. Ammonium, organic fractions (black carbon as well as organic carbon), and organic compounds like formate and vanillic acid are found to be good proxies for tracing past boreal fires in Greenland ice. We show that use of other species – potassium, nitrate, and carboxylates (except formate) – is complicated by either post-depositional effects or existence of large non-biomass-burning sources. The quality of levoglucosan with respect to other proxies is not addressed here because of the present lack of high-resolution profiles for this species, which does not allow for a fair comparison. Several Greenland ice records of ammonium consistently indicate changing fire activity in Canada in response to past climatic conditions that occurred during the last millennium and since the last large climatic transition. Based on this critical review, we make recommendations for further study to increase reliability of the reconstructed history of forest fires occurring in a given region.

## 1. Introduction

Biomass burning is a major source of gases and aerosols (Andreae and Merlet, 2001; Akagi et al., 2011) that strongly influence the chemical composition of the atmosphere and the radiation balance. In turn, climate changes directly disturb the fire regime – for instance, through the frequency of lightning ignition following enhanced summer convective activity (Romps et al., 2014)– including the occurrence and duration of fire weather (Soja et al., 2007). Climate-induced changes in vegetation (changing type of vegetation, flammability of



species, fuel mass) also may influence the fire regime (Hély et al., 2001). In this context, the boreal forest, a key carbon reservoir, is of particular interest because of the dominance of natural fires in this region where future warming would very significantly modify summer weather conditions. Whereas the burned forest area is expected to increase in the future in boreal regions (Flannigan et al., 2005), spatial and temporal differences in

the response since 1900 (Flannigan et al., 2001; Girardin et al., 2009) seem to have occurred. The recent availability of satellite data has strongly increased the accuracy of estimated burned areas (particularly for Siberian fires; Conard et al., 2002). Examination of present-day climate, fire conditions, and vegetation interactions, however, is complicated by the fact that, in addition to natural fires, many recent fires also are caused by human activities. Humans manage and extinguish fires, further complicating the spatial and temporal

record of fire activity. Thus, there is a need to examine climate, fire conditions, and vegetation interactions through time prior to the appearance of human-origin fires.

Charcoal deposited in lake and peat sediments is a generally accepted proxy for reconstructing fire occurrence in the past (Clark et al., 1996). Fire signals derived from charcoal analysis is dependent on the applied technique that either measures microscopic charcoal (<100 µm) in pollen-slides or macroscopic charcoal (>200 µm) by

sieving sediments (Carcaillet et al., 2002 and references therein). Whereas sieved macro-charcoal is representative of local-scale fire (typically a few hundred meters from the sampling site), micro-charcoal from pollen slides is considered a proxy of regional burning history (up to 1 km$^2$ around the sampling site; Carcaillet et al., 2001). Because of this limited spatial representativeness of charcoal records, several individual records must be assembled to derive a regional reconstruction of fire activity. In this way, paleo-fire records have been

obtained at regional, continental, and global scales. North American paleo-fire records covering the Holocene, for example, are available for four distinct regions (Northwestern boreal, St. Lawrence region, western U.S., and central North America) (Marlon et al., 2013).

Studies of the chemical composition of snow and ice have allowed examination of natural variability of the chemical composition of the past atmosphere (see Legrand and Mayewski (1997) for a review). Numerous

chemical species emitted or produced in combustion plumes are trapped in glacial archives. Thus, ice cores can be used to reconstruct past fire activity, providing a tool complementary to charcoal records with unrivalled dating precision in examining the link between climate and fire activity. We also emphasize that in high northern latitude regions, charcoal records do not extend prior to 12 kyrs BP since these regions were covered by ice. Potential fire proxies in ice are inorganic species (ammonium, nitrate, and potassium), as well as black carbon

and numerous organic carbon species (or groups of species) that also are produced during combustion. These latest species include carbohydrates such as monosaccharide anhydrides (levoglucosan), aldehydes, short-chain (C1-C5) mono- and di-carboxylic acids, and acidic oligomers (humic-like substances). In addition, other organic molecules are specifically emitted during combustion of a certain type of fuel: dehydroabietic acid produced by pyrolysis of conifer resin (Simoneit et al., 1993) and phenolic acids produced during incomplete combustion of

conifers (vanillic acid) and grass (p-hydroxybenzoic acid) (Simoneit et al., 2002).

The finding that ice cores archive forest fire plumes was first supported by Legrand et al. (1992) by showing that the concentration of ammonium, formate, and oxalate (measured by ion chromatography) in some summer layers deposited at Summit (central Greenland) greatly exceed their background levels. The authors attributed these ammonium/carboxylate events – mainly constituted by ammonium formate salt (ammonium to formate mass

ratio of 0.4) – to transport of forest fire plumes from North America to the Greenland ice cap. Although the



mechanism leading to formation of ammonium formate in biomass burning plumes was not elucidated at that time, the presence of micro-soot particles identified by electron microscopy in one of these events confirmed the biomass burning origin of these ammonium events (Legrand et al., 1995). This also was confirmed by atmospheric studies conducted at Summit showing a sudden increase of these chemical species in August 1994

and June 1993, when biomass burning plumes were transported to the site from northern Canada (Dibb et al., 1996; Jaffrezo et al., 1998). Finally, Kehrwald et al. (2012) showed that Summit snow layers corresponding to August 1994 Canadian forest fires exhibited in addition to an oxalate peak a large increase in levoglucosan, a carbohydrate specifically produced by cellulose burning.

Following these pioneering investigations, numerous studies have reported on past frequency of forest fires.

Most of them were conducted in Greenland snow and ice, and often used different chemical proxies measured at different time resolution (sub-annual to multiple years). To date, using high-resolution (sub-annual) ammonium records obtained at central Greenland sites through the continuous flow analysis (CFA) technique, Fuhrer et al. (1996) investigated biomass burning for part of the Holocene (from 7.9 to 11.6 kyrs BP and from 2.8 kyrs BP to present) at Summit and Fischer et al. (2015) during the last glacial cycle (from 10 to 110 kyrs BP) at North

GRIP. Since ammonium but not formate can be measured with CFA, high-resolution records of ammonium together with formate remain limited until now to the last millennium as obtained at Summit by using IC with sub-annual sampling (Legrand and De Angelis, 1996; Savarino and Legrand, 1998). Following the availability of black carbon (BC) analyzers (SP2, Droplet Measurement Technologies, Boulder, Colorado), high-resolution records became available. For instance, McConnell et al. (2007) reported the first high-resolution record of BC

in Greenland ice at the western central site of D4. Spanning the 1788 to 2002 time period, complemented by measurements of vanillic acid, this BC record permitted evaluation of the contribution of biomass burning and fossil fuel in Greenland snow deposited during the two last centuries. Alternatively, a low-resolution (5- to 50-year average) profile of levoglucosan covering the last 15 kyrs at the NEEM site in northwest Greenland was used by Zennaro et al. (2015) to investigate boreal fire activity across millennial timescales. None of these

studies, however, addressed the following critical questions. First, how consistent are the fire records obtained at different Greenland sites (northern versus central), and did atmospheric transport between the source regions and drill site influence the record? Since different fire proxies have different emission factors depending on the combustion conditions and fuel nature, different atmospheric lifetimes, and some are not exclusively emitted by combustion, a comparative study of the record of these different proxies at a given site would be more than

welcome to examine the quality of these different proxies. Finally, biomass burning ice records were gained at sites located in mid-latitude northern regions but they remain sparse and limited to the last centuries, see Eichler et al. (2011) for a Siberian Altai glacier (the last 750 yrs), Kawamura et al. (2012) for a Kamchatka glacier (the last 300 yrs), and Müller-Tautges et al. (2016) in the Swiss Alps (the last 40 yrs).

In this paper, we first review available ice core chemical records of biomass burning focusing mainly on

Greenland cores, discussing the quality of different proposed proxies considering the intensity of their emission factor during combustion, and their atmospheric behavior (lifetimes and budgets). We then compare several high-resolution (sub-annual) records extracted at different sites, scrutinizing recent decades (1950-1990) during which meteorological data are available for backward air mass trajectory calculations and burned boreal forest area became more accurately known. This permits exploration of boreal source regions influencing the

Greenland ice core records of forest fires and evaluation of the extent to which atmospheric transport influences



these records. Although not recorded in all Greenland ice cores, the origin of the 1908 event (the year of the Tunguska phenomenon) is discussed in detail. Some conflicting conclusions on long-term trends in boreal forest fires, in particular for the last millennium and last glacial-interglacial change, are discussed. Based on this critical review, we make recommendations for further studies to strengthen the quality of ice core records required to examine the relation between climate, vegetation, and fires in the past.

## 2. Previously published and unpublished data

Ice core studies from different Greenland sites that investigated past boreal forest fires are summarized in Table 1. From North to South, the documented sites include NEEM (77°26'N, 51°03'W), North GRIP (75°N 42°W), Summit (72°36'N, 38°18'W), D4 (71°24'N, 43°54'W), and 20 D (65°N 44°52'W). In addition to these published data, we used several unpublished seasonally resolved CFA records obtained by the Desert Research Institute (DRI) on firn and ice cores from some of the sites shown in Table 1 (NEEM-2011-S1, SUMMIT 2010, North GRIP 2, and D4), and from the Humboldt site located in NW Greenland (78°31'N, 56°49' W; 1985 m asl; accumulation of 14.4 g cm$^{-2}$ yr$^{-1}$). Other unpublished data used in this study were obtained in the field during the GRIP campaigns at Summit, including CFA measurements of ammonium and formaldehyde (Physics Institute and Oeschger Center for Climate Change Research at Bern), and IC investigations of short-chain mono and dicarboxylates (Laboratory of Glaciologie et Géophysique de l'Environnement at Grenoble).

Details on sampling and methods generally can be found in publications referenced in Table 1. Following the very first investigations made by Hagler et al. (2007) in snowpit samples collected at Summit, new investigations recently addressed the amount of organic carbon (OC) present in Greenland ice. Quantification of the dissolved organic carbon (DOC) content of ice on discrete samples was developed at LGGE using an ultraviolet (UV) oxidation and infrared (IR) quantification of $CO_2$ with a commercial Phoenix 8000 system (Teledyne Tekmar Company, CA). To remove contamination present on the outer portion of ice samples, a glass device was designed in which ice samples are washed with ultrapure water and melted afterwards (Preunkert et al., 2011). During sample melting, an inert gas atmosphere is maintained inside the glass device to prevent contact of the ice sample with ambient air in the laboratory. These working conditions allow analysis of small (5 mL) samples containing less than 10 ppb C (Preunkert et al., 2011). They are, however, tedious (15 min required for cleaning and analysis of one sample) and difficult to apply for high-resolution Greenland ice records. Very recent improvements in sample handling and other techniques in the CFA system at DRI enable arguably the first reliable high-resolution measurements of total organic carbon (TOC). In this method, a Sievers 900 analyser is coupled to an ice core melter and the constantly flowing sample stream (isolated from any interaction with laboratory air) is analysed for TOC within a few minutes of initial melting. TOC measurements are most reliable in ice cores drilled without use of organic drilling fluids and in samples from below the pore close-off depth where potential contamination from circulating modern air through the core is eliminated.

## 3. Chemical species emitted by combustion and fingerprint of forest fires in ice

### 3.1. Chemical composition of biomass burning plumes

For species identified in biomass burning plumes, we summarize their origin in the plant biomass and their different chemical formation pathways (Fig. 1). Typically, the plant biomass contains C (45%), O (45%), H





(6%), N (1.5%), K (1%), Ca (0.5%), S (0.1%), and Cl (0.01%) (Raven et al., 2007). At any time, a vegetation fire may proceed under both flaming and smoldering conditions. During flaming (1400 K) that dominates in the earlier phase of the fire, C, H, and N present in fuel are converted into highly oxidized simple molecules ($H_2O$, $CO_2$, $N_2$, and NO). Most BC also is produced at this stage. Later, smoldering conditions (below 500–700 K) tend

to become dominant, leading to emission of most of the CO, non-methane hydrocarbons (NMHCs), and primary OC aerosol. Concerning the OC fraction, identification of individual species in many studies often remained limited to short chain mono- and di-carboxylates that represent less than 10% of total OC mass (e.g., Ruellan et al., 1999). Following Simoneit et al. (1999) who pointed out levoglucosan as a tracer of cellulose in biomass burning, several studies conducted in combustion plumes extended OC speciation to this carbohydrate. To our

present knowledge, the most extended OC speciation was obtained for savannah fires (Gao et al., 2003) where, in addition to carbohydrates like levoglucosan and glucose, gluconic acid was tentatively identified as the most important carboxylate. At temperatures higher than 400°C, levoglucosan can be thermally destroyed into either smaller compounds producing various aldehydes and carboxylic acids or repolymerized into polysaccharides to then form high molecular organic containing bonds and carbonyl groups (i.e., similar to humic-like substances;

Fig. 1) (Kawamoto et al., 2003; Abella et al., 2007). Finally, in addition to these primary emissions, numerous NMHCs and oxygenated volatile organic compounds (OVOCs) emitted during combustion will be oxidized into small carboxylic acids (Fig. 1). In contrast to combustion of fossil fuels, the relatively low temperature of biomass burning compared to fossil fuel combustion leads to emission of nitrogen (mostly NO and $NH_3$) mainly originating as nitrogen present in fuel (not as atmospheric $N_2$). Potassium, a major electrolyte in plant cytoplasm,

is easily volatilized during combustion (boiling point of 760°C against 1484°C for calcium for instance).

As discussed above, emission of many species often depends on combustion conditions, particularly for organic compounds. The content of burned matter is another important parameter for many inorganic compounds. Quantification of emission factors, defined as the mass of species emitted per kg of dry matter (DM) burned, during combustion of different types of vegetation highlights some specific characteristics of boreal fires. For

most species reported in Fig. 1, emission factors can be found in Andreae and Merlet (2001) for savannah as well as tropical and extratropical forests (including boreal and temperate fires). More recently, data have become available for some species distinguishing between temperate and boreal fires (Akagi et al., 2011). One of the most important emission features from boreal fires lies in nitrogen emissions dominated by ammonia but not $NO_x$ emissions. The case of nitrogen emissions is a good example of the need to get data distinguishing

temperate from boreal fires. Indeed, whereas Andreae and Merlet (2001) reported an emission factor twice higher for NO than for $NH_3$ (3 g of NO against 1.4 g of $NH_3$ per kg of DM) for extratropical fires, Akagi et al. (2001) report emission factors of 2.7 g for $NH_3$ against 0.9 g for NO per kg of DM for boreal fires (0.8 g of $NH_3$ against 2.5 g for NO per kg of DM for temperate fires). For potassium, although no data are available for boreal fires, Andreae and Merlet (2001) reported emission factors ranging between 0.1 and 0.4 g of potassium per kg of

DM for extratropical fires. Potassium emission factors are not expected to be strongly dependant on fire combustion (McMeeking et al., 2009) but more on potassium content of the fuel. For organic compounds (NMHCs, OVOCs, and OC primary aerosol), smoldering conditions that are predominant in boreal fires generally lead to higher emission factors than other vegetation fires. Only available for boreal fires are emission factors for NMHCs and OVOCs (a total of 59 g against 24 g kg$^{-1}$ of DM for temperate fires; Akagi et al. (2011)).

These emissions of NMHCs and OVOCs play a key role as precursors of short chain carboxylates, representing a





large source of secondary OC aerosol (Fig. 1). For carbonaceous aerosol (BC and OC), laboratory experiments by McMeeking et al. (2009) show that, as expected, twice more BC is emitted under flaming conditions than under mixed conditions (flaming and smoldering) (1.0 g of BC per kg of DM for flaming and 0.5 g of BC per kg of DM for mixed conditions). Alternatively, far less OC is emitted under flaming conditions (1.6 g of OC per kg of DM for flaming and 16 g of OC per kg of DM for mixed conditions). Finally for levoglucosan, although less data are available, Andreae and Merlet (2001) reported emission factors of 0.75 g of levoglucosan per kg of DM for extratropical fires (against 0.28 for savannah). The expected dependency of emission factors of levoglucosan with combustion conditions characterized by a quasi-zero emission during flaming, as observed by Gao et al. (2003) for savannah fires, was confirmed by laboratory experiments (Dhammapala et al., 2007; Kuo et al., 2008).

### 3.2. Fingerprint of forest fires in Greenland ice

As seen in Table 1, with the aim to reconstruct past forest fire activity, ammonium has been extensively used to document fires in numerous Greenland ice cores at sub-annual resolution. Many of these studies also documented nitrate and more occasionally soluble potassium deposition. These high-resolution studies sometimes documented BC and some organic acids (particularly C1-C2 carboxylic, and vanillic acid). Other Greenland records exist for levoglucosan but only at low resolution (Table 1). Since fires are episodic in nature, we first focus the discussion on ammonium, short chain carboxylates, BC, nitrate, and potassium for which several high-resolution Greenland profiles are available. The case of levoglucosan will be discussed later in Sect. 4.2 and 4.3.

As seen in Table 1, high-resolution chemical profiles were obtained in Greenland ice cores by employing either IC analysis on discrete samples or CFA techniques. The first records were achieved by subsampling pieces of firn and ice (several samples per year) with subsequent analysis by IC that permits investigations of potentially relevant species like ammonium, potassium, nitrate, and several light carboxylates (formate, acetate, glycolate, and oxalate). The second set of profiles were achieved using CFA – permitting investigation of high-resolution records of ammonium, nitrate, formaldehyde, BC, and sometimes vanillic acid.

As discussed by Legrand and De Angelis (1996), the firn material for carboxylates is very sensitive to contamination during standard storage of cores in sealed plastic bags. To minimize this problem, a firn core (73 m depth) drilled at Summit in 1993 (Legrand and De Angelis, 1996) was processed in the field by shaving around 90% of the outer part of firn core sections within one or two hours after extraction. Note that the obtained profile still represents the single profile documenting carboxylates during recent decades in Greenland snow. In order to examine the chemical signature of ammonium events with respect to background levels, data gained along this firn core were used to examine the relationship between ammonium and nitrate as well as between ammonium and the different carboxylates (Fig. 2). Anthropogenic emissions having impacted the budget of nitrate and carboxylates in Summit snow layers, we restricted our examination to the pre-1940 time period (520 samples). Indeed, the level of nitrate between 1250 and 1940 AD ($60 \pm 17$ ppb; Savarino and Legrand, 1998) was enhanced to $99 \pm 37$ ppb during recent decades as a result of growing $NO_x$ emissions (Mayewski et al., 1990; Fischer et al., 1998). Furthermore, Legrand and De Angelis (1996) demonstrated that the levels of formate close to $9 \pm 3$ ppb between 1750 and 1940 have decreased to $6 \pm 2$ ppb as a result of acidification of the atmosphere following emission of $SO_2$ and $NO_x$. In Fig. 2, we also used data obtained along pieces of ice (660



samples) from the deep GRIP ice core and corresponding to the Holocene period (Legrand and De Angelis, 1995, 1996).

### 3.2.1. Ammonium and Carboxylates

As shown in Fig. 3, the ammonium peaks are accompanied by a simultaneous increase of formate, oxalate, and
sometimes nitrate and potassium. For glycolate and acetate, the snow or ice layers located in the vicinity of the ammonium peak also are disturbed. The presence of these wider peaks for glycolate and acetate than for ammonium likely are related to post- depositional processes acting on these two volatile species. A similar smoothing of the HCHO input related to biomass burning events also was reported by Fuhrer et al. (1993), as seen in the example shown in Fig. 4. Therefore, in examining the relationship between ammonium and
carboxylates in Fig. 2, we used glycolate and acetate concentrations in samples corresponding to the background and peak but discarded those in the neighboring layers of the ammonium peaks (i.e., 27 values of 524 during the 1773-1940 time period and 30 values of 630 during the Holocene). Fig. 2 shows that formate unambiguously increases well above its background level during ammonium events. The ammonium events also are accompanied by enhancement above background values for glycolate, and to a lesser extent oxalate, whereas for
acetate the enhancement remains within the range of background values variability (Fig. 2).

To compare deposition of carbon associated along ammonium events, we calculate in Fig. 5 enhancement of the different carboxylates above their respective background values for different ammonium events. For species exhibiting broadening with respect to the ammonium peak (acetate and glycolate), we considered total deposition by calculating the total area of the perturbation. The same procedure was applied to formaldehyde
data available along a few ammonium events, such those reported in Fig. 4. Using the slope of the linear regression between ammonium and carboxylates (expressed in carbon mass) plotted in Fig. 5 (formate: $[HCOO^-]$ = 0.65 $[NH_4^+]$ $R^2$ = 0.86, acetate: $[CH_3COO^-]$ = 0.07 $[NH_4^+]$ $R^2$ = 0.32, glycolate: $[CH_2OHCOO^-]$ = 0.105 $[NH_4^+]$ $R^2$ = 0.69, and oxalate: $[C_2O_4^{2-}]$ = 0.025 $[NH_4^+]$ $R^2$ = 0.45), and the one calculated for formaldehyde ($[HCHO]$ = 0.02 $[NH_4^+]$ $R^2$ = 0.86, not shown), we estimate that input of formate accounts alone for 74% of the total mass of
identified OC species followed by glycolic acid (12%), and acetic acid (8%), HCHO and oxalate accounting each for 3%. This dominant presence of formate in ammonium-rich Greenland deposits contrasts with the relatively weak emission factors reported for boreal fires by Akagi et al. (2011) (0.16 gC for formic acid against 1.8 g C for acetic acid per kg of DM). As discussed by Lefer et al. (1994), this reflects the importance of secondary production of formic acid during aging of the plume.
Fig. 5 includes a few data corresponding to ammonium events that occurred during the Younger Dryas (11,600-12,600 yrs BP); these are reported as triangles. Of the four events reported in the formate panel, we were able to estimate enhancement above background levels for only one event because of lack of available ice samples adjacent to the event. As seen in Fig. 5, this fully documented Younger Dryas event the ammonium peak of which reaches 145 ppb, indicates no enhancement of glycolate. In fact, in the deep GRIP ice core, the
background level of this carboxylate gradually decreases through time, from 1.3 ± 2.6 ppb during the last 5 kyrs to 0.3 ± 0.9 ppb between 5 and 10 kyrs BP and zero prior to 11 kyrs BP (not shown). This decreasing trend is not related to past climate changes since, even during the preceding warm period (the Eemian at around 120,000 yrs BP), glycolate was still not detected in ice whereas all other carboxylates are in the range of values seen during the Holocene. We therefore have to conclude that this species slowly degraded in the ice.



Although present at concentrations well below those of above discussed organic compounds (less than 1 ppb), vanillic acid was investigated using CFA in Greenland snow layers (e.g, at D4; McConnell et al., 2007) with the aim of attributing the contributions of fossil fuel and biomass burning to the budget of BC during the two last centuries. In Fig. 6 we report annual levels of vanillic acid and ammonium in snow layers deposited at D4 between 1740 and 1870. It can be seen that vanillic acid also can be used as a surrogate of ammonium ($R^2 = 0.62$ for vanillic acid versus ammonium, compared to $R^2 = 0.87$ for formate versus ammonium at Summit).

### 3.2.2. Ammonium versus Formate and Nitrate

Whitlow et al. (1994) and Savarino and Legrand (1998) pointed out that some (but not all) $NH_4^+$ events in the Summit layers are accompanied by a moderate increase (a few tens of ppb) of nitrate. Fig. 3 illustrates this with the 1863 ammonium event showing an increase of nitrate (~40–50 ppb) but not the 1908 event. Fig. 2 suggests that, if it exists, an input of nitrate accompanying ammonium events remains in the range of variability of nitrate background levels. The variability of nitrate background level in Summit snow layers is mainly related to the seasonal change with values closed to 50 ppb in winter and 90 ppb in summer. Although the budget of nitrate in pre-industrial Greenland ice has not been discussed yet, it is likely that the summer maximum is related to natural NO sources like soil emissions from surrounding continents and possibly lightning, both being more active in summer than in winter. Fig. 2 does not support use of nitrate in Greenland ice as a proxy of forest fires. While Legrand and De Angelis (1996) reported a formate to ammonium molar ratio close to unit along $NH_4^+$ events recorded at Summit both during the last 200 years and the Holocene, Savarino and Legrand (1998) showed that from 1190 to 1770, more ammonium events have been accompanied by a significant increase of nitrate. The input of nitrate plus formate equilibrates the ammonium input on a molar basis. As seen in Fig. 7, the departure from unit of the ammonium to formate molar ratio in snow remains more an exception than an overall rule. With an emission factor of 2.7 g of $NH_3$ against 0.9 g for NO per kg of DM in boreal fire emissions (see Sect. 3.1.), we would expect ammonium to dominate nitrate by a factor of five on a molar basis. The even larger observed dominance of ammonium in forest fire debris deposited in Greenland snow suggests that, in addition to emission factors of the two species, another parameter tends to minimize deposition of nitrate. Aircraft-based sampling completed during the NASA Arctic Boundary Layer Expedition (ABLE 3B) indicated an $HNO_3$ mixing ratio of 200-400 pptv against 1300-3000 pptv of formic acid and 1700–3000 pptv of acetic acid (Lefer et al., 1994). A detailed study of $NO_y$ partitioning in boreal biomass burning plumes was recently completed by Alvarado et al. (2010) in the frame of Artic Research of the Composition of the Troposphere from Aircraft and Satellite (ARCTAS-B), showing that PAN is the dominant $NO_y$ species well above $HNO_3$. We also emphasize that the relative abundance of NO with respect to $NH_3$ would depend on the altitude of the plume. It is expected that most NO emissions take place during flaming when the release of energy is at its maximum, sometimes permitting plumes to reach the upper troposphere by pyro-convection. These events are rare, and boreal fire plumes are generally observed between 2 and 7 km (de Gouw et al., 2006).

Using the tropospheric emission spectrometer aboard the NASA Aura satellite, Alvarado et al. (2011) reported $NH_3$ mixing ratios of several ppbv (up to 7 ppbv) in fresh biomass burning plumes in July 2008 over central Canada. Similarly, HCOOH mixing ratios ranging from 1 to 2.4 ppbv were observed. Given the dominant presence of these two gases in these boreal fire plumes, we may expect that – during transport toward Greenland – a neutralization of ammonia by formic acid would take place through in cloud processes. Since satellite data





indicate that ammonia is present in excess with respect to acidic species, it is likely that the neutralization process can continue until the secondary production of HCOOH from the oxidation of NMHCs and OVOCs stops. This may explain the quasi-invariant presence of ammonium formate in Greenland snow and ice layers impacted by boreal forest fire plumes.

### 3.2.3. Ammonium versus potassium

In the atmosphere, the fine fraction of soluble potassium is thought to be related to biomass combustion (Cachier et al., 1991). In the ice, this fraction can be estimated by subtracting both the contribution of sea salt by using sodium level ($[K]_{sea-salt}$ = 0.038 [Na]) and of dust by using calcium from the potassium present in ice (and measured with IC). For the latter contribution, Legrand and De Angelis (1996) derived a $K^+/Ca^{2+}$ mass ratio of 0.04 based on the observed linear correlation in snow layers free of biomass burning inputs along the GRIP 1993 firn core. In Fig. 3, the fine fraction calculated on this basis (i.e., $[K^+]_{bb}$ = $[K^+]$ - 0.038 * $[Na^+]$ - 0.04 * $[Ca^{2+}]$) is reported for the ammonium events dated at Summit in 1908 and 1863. While in the case of the 1863 event an increase of potassium in the range of 2 ppb is detected, no significant change is detected along the 1908 event. Examination of the relationship between fine potassium and ammonium along ammonium events having occurred between 1770 and 1890 confirms such a relatively poor relationship ($[K^+]_{bb}$ = 0.15 +0.03* $[NH_4^+]$ with $R^2$ = 0.3). Furthermore, even along the largest ammonium event of the Holocene ($NH_4^+$ concentrations reaching 410 ppb, Fig. 2), enhancement of potassium remained limited to 2 ppb (not shown) instead of 12 ppb if the preceding linear relationship is considered. Data from Summit therefore suggest that soluble potassium concentrations in snow and ice are enhanced by a few ppb along some (but not all) ammonium events. Given the high sensitivity of this species to contamination (Savarino and Legrand, 1998), it remains difficult to promote use of potassium as proxy of boreal fires.

### 3.2.4. Ammonium versus BC and OC

As seen in Fig. 6, the correlation between ammonium and BC remains relatively strong ($R^2$ = 0.69), although slightly weaker than the one between ammonium and formate ($R^2$ = 0.87 at Summit). At least for the Greenland sites discussed in this paper, the ammonium formate peaks always stay in summer snow layers without spreading to the rest of the year, in contrast to what is observed for formaldehyde (HCHO) and acetate (Sect. 3.2.1). We can, therefore, rule out that remobilization of ammonium formate after deposition (not foreseen for the irreversibly trapped BC species) renders the relationship between ammonium and BC weaker than the one between ammonium and formate. The most likely cause of the slightly weaker BC-$NH_4^+$ relationship compared to that of formate and ammonium is that flaming (the most important combustion phase for BC) is not the dominant phase in boreal fires, rendering the amount of BC reaching the Greenland ice cap variable from one event to another.

Apart from a snowpit study of recent snow layers deposited at Summit (Hagler et al., 2007), data on the OC content of Greenland ice are very rare. A few investigations of the amount of OC present in Greenland ice were first accomplished using an UV oxidation and IR quantification of $CO_2$ (Preunkert et al., 2011). Deployed for the examination of an ammonium event in the NEEM 2011-S1 core, Legrand et al. (2013) showed that the OC background level close to 15-20 ppb C was enhanced by ~ 100 ppb C along the ammonium event. As seen in





Fig. 8, a very similar picture was obtained at DRI using CFA coupled with a TOC analyser. Although limited, these data highlight two important points. First, the carbon input related to biomass burning is more than a factor of 10 higher as OC than as BC. Laboratory experiments by McMeeking et al. (2009) show that, under mixed conditions (flaming and smoldering), at the emission stage, the OC/BC ratio would exceed 10. Consistency

between these two numbers, however, is likely coincidental since BC would have a longer atmospheric lifetime than OC but secondary OC production likely counteracts the larger sink of OC with respect to BC. The second important information in Fig. 8 is that, given the main fingerprint of fire in Greenland snow (ammonium formate), a large part of the OC enhancement in ice layers containing biomass burning material is due to the presence of formate. Therefore, the possibility to obtain a high-resolution TOC profile would offer an excellent

complementary tool to ammonium CFA measurements to extract relevant high-resolution biomass burning records from Greenland ice.

### 3.2.5. Levoglucosan

Levoglucosan is exclusively produced by fuel combustion and formed in large quantities by combustion of cellulose (Simoneit et al., 1999) during smoldering. It therefore has been widely used as tracer of biomass

burning. This is even more interesting for boreal fires that are dominated by smoldering. Note, however, that as shown by Kuo et al. (2008), the formation of levoglucosan is highly sensitive to the combustion temperature with a yield reaching a maximum around 250°C. Furthermore, several recent laboratory studies have highlighted the existence of several degradation pathways that may be efficient under atmospheric conditions. They involve heterogeneous oxidation by gaseous oxidants (Lai et al., 2014; Henningan et al., 2010, Sang et al., 2016), gas

phase oxidation of gaseous levoglucosan (May et al., 2012), and aqueous phase degradation (Holmes et al., 2006; Zhao et al., 2014). These studies proposed an atmospheric lifetime of a few days. It is never easy to extrapolate kinetic rates to conditions encountered in the real atmosphere, but a significant degradation of levoglucosan during transport cannot be ruled out, given the typical atmospheric transport time of three-to-ten days for boreal fire plumes to reach Greenland (Sect. 4.1.1). Further studies of Greenland ice can help to answer

this question, by comparing high-resolution records of levoglucosan with those of other proxies. Also, more atmospheric measurements of levoglucosan in fresh and aged boreal fire plumes would be useful.

### 3.2.6 Quality of the different proxies

At a given site, the importance of non-biomass-burning sources with respect to the perturbation associated with

arrival of a biomass burning plume influences the ability of a chemical species to represent a good proxy of fire in ice. The magnitude of the forest fire perturbation depends on the strength of emission and atmospheric lifetime of the considered species. As discussed above for Greenland ice, ammonium, formate, OC (DOC or TOC), BC, vanillic and glycolic acids were enhanced well above their background values. Levoglucosan also would stay in this group of species (denoted group 1) for which non-biomass-burning sources are relatively weak

or do not exist. HCHO also would in group 1 since biomass burning is expected to represent an important source in the free troposphere of remote regions where its main source is methane oxidation (Lowe and Schmidt, 1983). Its use as fire proxy in ice, however, is strongly hampered by large post-depositional effects (see Fig. 4).





In contrast to the preceding species, other significant non-biogenic sources exist for soluble potassium (sea-salt and leachable potassium in dust), nitrate (NO soil emissions and/or lightning), and acetate (vegetation emissions and possibly marine biogenic emissions, Legrand and De Angelis, 1996).

Several other factors have to be considered in ranking the different proxies. In group 1, most species can be used whatever is the studied time period, except BC for which, as discussed by McConnell et al. (2007), anthropogenic emissions have grown since 1850. Since glycolate slowly disappears from ice with time (Sect. 3.2.1), its use as fire proxy is restricted to recent times. Another important consideration when evaluating the ability of different chemical species to trace boreal fire in Greenland ice is the performance of available analytical methods to obtain high-resolution, contamination-free profiles. In group 1, only ammonium, BC, OC, and vanillic acid can presently measured with CFA. It has to be emphasized, however, that contamination-free measurement of OC in porous firn material (i.e., corresponding to the two last centuries in Greenland snow layers) is very difficult (Legrand et al., 2013). Though being less dramatic than for OC, contamination of firn by formic acid during firn core storage in sealed plastic bags is also a difficult problem.

### 3.3. Fingerprint of forest fires in mid-latitude, high-elevation glacier ice

A few attempts were previously made to reconstruct past biomass burning from ice cores extracted at cold glaciers located at mid-northern latitudes. One factor is proximity to the site of permanent large biogenic emissions making use of some proxies that were tested in Greenland difficult (impossible, in some cases). One exception is the case of Mt Logan (60° 35' N, 140° 35' W, 5346 m asl) located in Yukon (Canada) where ammonium background values are still as low as those observed in Greenland, suggesting a relatively weak contribution of non-biomass burning sources in this region. Although no analysis of air mass origin was done, on the basis of poor agreement in the occurrence of ammonium events recorded at Summit and at Mt Logan during the last 250 years, Whitlow et al. (1994) suggested that this site is influenced by the Siberian fires in contrast to Summit more influenced by Canadian fires.

At lower latitudes, the impact of biogenic sources becomes more important than in Arctic regions. For instance, applying Principal Component Analysis (PCA) to major ions trapped in an ice core extracted from the Belukha glacier in the Siberian Altai mountains (49° 48'N, 86° 34'E, 4062 m asl) covering the last 750 years, Eichler et al. (2011) suggested that ammonium and formate mainly originate from biogenic emissions, and that only soluble potassium and nitrate in ice seem to represent good proxies of biomass burning for this region. The importance of biogenic emissions at that site also is evidenced by background values of ammonium and formate being one order of magnitude higher than those in Greenland (140 ppb of ammonium and 200 ppb of formate, Olivier et al. (2006)). It also has to be emphasized that the PCA analysis from Eichler et al. (2011) indicates that, as expected, potassium and nitrate are influenced by large emissions of dust from the desert of Central Asia.

Another site of interest to trace boreal fires from Siberia is located at Ushkovsky (56° 04' N, 160° 28' E, 3903 m asl) on the Kamchatka Peninsula, as shown by calculations of air mass back trajectories (Kawamura et al., 2012). At that site, data on major ions (ammonium, formate, and nitrate) are not available but we can suspect that their use as fire proxies are even more complicated than in Altai. Indeed, the quasi-continuous volcanic emissions that occur in this region may strongly influence deposition of these ions at that site. Instead, more extended use of several organic fire proxies – levoglucosan, vanillic acid, p-hydroxybenzoic and dehydroabietic acids – was done





on this ice core (Kawamura et al., 2012), showing some interesting differences both in the occurrence of peaks and the trend in background values. In particular, it is suggested that the levoglucosan signal is related biomass burning from a wider region (Siberia, Far East, Northeast Asia and Southeast Asia) than the one (Siberia) influencing other organics like dehydroabietic (a specific tracer of pyrolysis of conifer resin).

## 4. Comparison of high-resolution records of biomass burning proxies in Greenland ice

In this section we examine the consistency of different records of boreal fires archived in Greenland ice across different timescales, starting with the most recent times for which more observations are available.

### 4.1 Last two centuries

In Fig. 9, we compare five ammonium high-resolution records spanning the two last centuries, two from
Summit, one from D4, and two from NEEM. In general, the ammonium peaks tend to be better marked at NEEM than at Summit. The two sites have similar snow accumulation rates, but summer precipitation is more pronounced at NEEM than at Summit (Steen-Larsen et al., 2011). Accordingly, this difference tends to suggest a higher likelihood of a fire imprint in wet deposition at NEEM. Furthermore, as discussed in Sect. 4.1.1, there is more often a direct atmospheric transport from the fire areas to NEEM than to Summit.

Whereas a peak-to-peak comparison is rarely perfect (matches are denoted with a dashed line in Fig. 9), in numerous cases the two sites reveal similar events within a departure of one or two years (i.e., within the dating errors individual cores). It seems likely that many of these are actually in the same year and reflect a slight dating error. Note that, even for ice cores drilled in a distance of a few meters at the same site, glaciological noise by wind reworking is high and may lead to a differing imprint of the same event in the snowpack.

Accordingly, based on a systematic shallow firn core study on aerosol deposition at the NEEM site, Gfeller et al. (2014) were able to show that for ammonium a single core archives only 70-80% of the interannual variability in the atmospheric aerosol load, while the rest is obliterated by wind reworking. This finding may explain the differences seen between the two NEEM records in 1921, 1973 and 1980 (Fig. 9).

During the two last centuries, the consistency is excellent between Summit and D4 whereas some differences
appear between Summit and NEEM. The most striking are the absence of record of the 1908 event at NEEM, and the weakness of the 1863 and 1961 events at NEEM compared to Summit. Conversely, the 1980-1981 events are better detected at NEEM (at least in the 2011 S1 core) than at Summit. The largest discrepancies between Summit and NEEM (1908 and 1961) will be discussed in more detail in the two following sections, starting with the most recent 1961 event.

### 4.1.1. Source Regions and Atmospheric Transport

For the most recent years, it is possible to examine to the extent to which air mass transport influences the record of fire events at a given Greenland site. The main boreal fire sources are Siberia and Alaska/Canada. Data on area burned in Canada are available since 1920 (Van Wagner, 1988) and became more accurate after 1959 with the Canadian National fire database (Canadian Forest Service, 2015, National Fire Database – Agency Fire Data.
Natural Resources Canada, Canadian Forest Service, Northern Forestry Centre, Edmonton, Alberta. http://cwfis.cfs.nrcan.gc.ca/ha/nfdb.) providing precise fire location, start date, and final size (Stocks et al., 2003). Far less precisely known is area burned in Siberia prior to 1997, when satellite data became available.





Prior 1997, only 1915 is known to have been a year of very large fires in Siberia. As discussed below, however, the atmospheric transport greatly favors Canada as the main source for Greenland.

Area burned in summer (May to August) in Canada from 1959 to 1992 is reported in Fig. 10, distinguishing between western provinces located west of 96°W (British Columbia, Yukon, Alberta, Northwest Territories,
Saskatchewan, Manitoba, and Wood Buffalo National Park) and East (Ontario, Quebec, Newfoundland and Labrador). For these 34 years, maximum area burned occurred in June and July except for 1968, 1977, 1986 (May), and 1981 (August). The increasing area during recent years concerned more western than eastern provinces (Stocks et al., 2003). As seen in Fig. 10, the total area burned was high in 1961 (3.8 Mha), 1976 (2 Mha), 1980-1981, and 1989 (more than 4 Mha). Prior 1959, the total area burned (not shown) was also high in
1923 and 1929 (3 Mha), 1937 (2 Mha) and 1941 (2 Mha) (Van Wagner, 1998). Considering a possible dating error of one to two years, the Greenland ice records consistently indicate ammonium events in 1923, 1929, 1937, 1941, and 1961 (mainly at Summit and D4), and 1980-81. The relationship between ammonium deposition in Greenland ice and area burned in Canada is poor, however. For instance, an ammonium event was detected at all sites in 1950, a year during which only 1.2 Mha burned in Canada. In contrast, during 1981 during in which a
maximum of 5.6 Mha burned, only one ice core (NEEM S1, Fig. 9) reveals an outstanding ammonium peak.

Backward trajectories have been computed using the Hybrid Single-Particle Lagrangian Integrated Trajectory model (Stein et al. 2015, available at: http://ready.arl.noaa.gov/HYSPLIT.php) with meteorological data from NCEP/NCAR. Reanalysis data are available since 1948 (https://ready.arl.noaa.gov/archives.php). The model was run every 6 h in backward mode for three different altitudes (0, 250, and 500 m above ground level) for
May/June/July/August when, as discussed above, the fire activity in Canada (Siberia as well: van der Werf et al. (2006)) reached its maximum. Backward air mass trajectory calculations documented the time that air masses reaching Greenland sites were over the forested area of Canada, distinguishing between western and eastern provinces, and north of 45°N in Eurasia, distinguishing between western Europe (from 7 to 45°E) and Siberia (from 45 to 164°E). Comparison of trajectories arriving at each site at 0, 250 m, and 500 m above ground level,
revealed no significant difference. In the following, we report results obtained at 500 m above ground level. Typical backward trajectories calculated for five and ten days for the Summit and NEEM sites (500 m agl) are reported for 1978 in Fig. 11, indicating transport that tends to be slower at NEEM than at Summit. As a consequence, considering ten-day backward trajectories, on average in June-July 1978, the time spent over Canada by air masses reaching Greenland was limited to 5.4% at NEEM compared to 10.8% at Summit (not
shown). This compared to 4.2% at NEEM and 8.2% at Summit across 25 years (Table 2). Fig. 11 also indicates a larger fraction of time spent over eastern than western Canada at both sites, consistent with the 25-yr averaged values reported in Table 2. For instance, on average the ten-day backward trajectories arriving at NEEM spent 2.6% of time over eastern Canada against 1.6% over western Canada (4.6% over eastern Canada against 3.0% over western Canada at Summit). Given the fact that on average in June and July, only in a quarter of cases air
mass arriving at Summit (40% at NEEM) had travelled over North America, over June-July during a total of 4.3 days the five-day backward trajectories arriving at Summit had spent at least one day over forested areas of Canada (1.4 days at NEEM). For ten-day backward trajectories, the number of days during which the air masses spent more than one day reaches 15 days at Summit and 9 days at NEEM. At both sites, when air masses are not coming from North America, they have mainly travelled over the Atlantic Ocean and only rarely (3% to 10% of
cases) had travelled over Eurasia. This summer dominance of air masses coming from North America instead of

Eurasia reaching Summit is consistent with other studies (Kahl et al., 1997; Gfeller, 2015). As a result, even for ten-day backward trajectories, the time spent over the fire region of Siberia is 1% at Summit and 0.4% at NEEM (Table 2). Interestingly, as shown in Table 2, there is no significant change of air mass origin for summers during which large fires took place in Canada. From this multiple summer analysis of air masses reaching NEEM and Summit sites, we conclude that for species having atmospheric lifetime of a few days boreal regions of Canada (in particular Eastern provinces) are the main source regions for fire signals archived in Greenland ice.

We now focus on the 1961 event that is well recorded in central Greenland but missed in North Greenland snow layers. With a total area burned of 2.8 Mha (maximum in June with contributions from both western and eastern provinces), 1961 is clearly a particularly high fire activity year of the 1960s and 1970s. In Fig. 12, we reported the starting date of large fires (>200,000 ha) having occurred in June and July 1961 together with the fraction of time spent by air masses over western and eastern provinces of Canada reaching the two Greenland sites. Most of the area burned and fire emissions take place during the first four to eight days after the start of the fire (Mike Flannigan, personal communication 2016), Fig. 12 obviously shows that air masses reaching NEEM had far less sampled 1961 biomass burning plumes than those arriving at Summit. This example suggests that, in addition to the occurrence of snowfall, atmospheric transport also can modulate the magnitude of deposition of fire debris from site to site in Greenland ice.

### 4.1.2 The particular 1908 Event

Whereas the different high-resolution ammonium profiles obtained in various Greenland ice cores covering the last century (Sect. 4.1) are generally in reasonably good agreement, there are some exceptions, including the 1908 summer. Indeed, on the 30[th] of June 1908 anomalous optical phenomena were observed in the sky over Eurasia. These observations were attributed to the explosion of a cosmic body above the region of the Tunguska River basin in central Siberia (60°55'N, 101°57'E). A review of this event can be found in Hugues (1976). This event has been the subject of debate regarding several aspects, including the nature of the bolide – a low-density cometary body or stony meteorite. In the first case, accompanying the progressive ablation of the body during its entrance into the atmosphere, thermal decomposition of $N_2$ and $O_2$ would have efficiently produced NO through the mesosphere and stratosphere (Turco et al., 1982). Estimation of the production reached 30 Tg of NO and would be detected as enhanced nitrate levels in snow deposited in both Greenland and Antarctica during at least one year. Neither from Greenland (Camp Century and Dye 3; Rasmussen et al., 1984) nor from various Antarctic sites (Dome C; Legrand and Delmas, 1986; South Pole; Legrand and Kirchner, 1990), was such an increase of nitrate in ice detected. Based on a more accurate aerodynamic model of the bolide, Chyba et al. (1993) suggested that Tunguska was a stony body, which exploded at an altitude of 9 km with all the energy deposited between 12 and 5 km altitude. On this basis, Curci et al. (2004) calculated that 0.4 Tg of NO were produced within this elevation range and that nitric acid was mainly deposited downwind within the first month.

Another debate concerned the possibility that a fragment of the Tunguska asteroid survived strong heating within the atmosphere and collided with the ground. For instance, Gasperini et al. (2007) suggested that Lake Cheko, a 300 m wide lake located a few km from the epicenter of the 1908 Tunguska event, is an impact crater. Based on observational evidence that contradicts this hypothesis (no evidence of an uplifted rim as seen at many fresh impact craters, the presence of unaffected mature trees close to the lake), Collins et al. (2008) concluded that



Lake Cheko is highly unlikely to be an impact crater. This debate is of importance with respect to the possibility raised by Melott et al. (2010) that the Tunguska event also may have significantly produced ammonia from nitrogen and hydrogen (the so-called 'Haber-like process") under high pressure in the shock front of the bolide. Indeed, such ammonia production needs a sufficient amount of water to proceed and therefore requires the event

to have been accompanied by a ground impact. We will further discuss this point below. There is a consensus that the shock wave of the Tunguska explosion devastated trees across an area 0.215 Mha (Fast et al., 1967), and the flash burned vegetation covered an area of about 0.01 Mha. Abramov et al. (2003) estimated that following this event, a total of 0.09 Mha were burned. This burned area remains one or two orders of magnitude lower than the annual area burned in North America, however (2 Mha on average, up to 7 Mha during some years; Stocks et

al., 2003) and in Eurasia (12 Mha on average; Conard et al. 2002).

Fig. 13 summarizes sub-annual-resolution $NH_4^+$ records from several Greenland ice cores spanning 1905 to 1916. It appears that, for ammonium, 1908 stands out at Summit and D4 but not at NEEM and Humboldt. As discussed in Sect. 3.2.2, an increase of nitrate is sometimes detected along with the ammonium peaks at Summit, but when present, they rarely exceeded a few tens of ppb. As seen in Fig. 13, neither at D4 nor at Summit the

ammonium event was accompanied by a significant enhancement of nitrate. In contrast to the absence of nitrate peak at Camp Century and Dye 3, two other studies conducted at Summit reported a large nitrate peak in the 1908 layer. McCracken et al. (2001) reported the presence of two narrow sharp spikes (one of them reaching 180 ppb above the usual background in summer) in the nitrate record during 1908, no other relevant species being reported in their study. Melott et al. (2010) reported unpublished data from Summit showing an ammonium peak

(200 ppb above the usual background winter level) together with a nitrate peak (around 150 ppb above the usual background winter level) during winter 1908/1909. Both of these studies assigned this ammonium to the Tunguska event. Since the 1908 ammonium event detected in Greenland has a very similar organic signature (an input of ammonium formate at Summit (Fig. 5), and the presence of vanillic acid at D4 (Fig. 6)) compared to other events, we can rule out the hypothesis that an event like Tunguska was able to produce ammonium

following the Haber reaction as proposed by Melott et al. (2010). We therefore do not agree with the conclusion from Zennaro et al. (2014) who explained that the absence of levoglucosan and ammonium spikes in the 1908 NEEM ice core suggest that ammonium peaks in other Greenland ice cores may not be caused by forest fires during the 1908 summer but by a Haber-like process as invoked by Melott et al. (2010). Instead, as discussed in Sect. 4.1, the lack of a pronounced 1908 peak at NEEM is likely due to the local character of fire plume transport

to and/or deposition onto the Greenland ice sheet.

Another point concerning the 1908 event may come from studies conducted in Eurasia. Kawamura et al. (2012) measured biomass burning organic tracers in an ice core spanning 300 years extracted from Ushkovsky ice cap. Although discontinuous, the levoglucosan record showed sporadic peaks in 1705, 1759, 1883, 1915, 1949, and 1972, with the largest peak in 1949. Similarly, vanillic and p-hydroxybenzoic acids exhibited peaks in 1705,

1759, and 1949. Although estimates of burned area in Siberia are likely greatly underestimated, the 1915 year was well recognized as a dry year during which a total of 140,000 $km^2$ were estimated to have burned in Siberia (Shostakovitch, 1925). Since the high levels of levoglucosan in the 1915 layer of the Ushkovsky ice core coincides very well with the year of large fires in Siberia and none of the measured biomass burning organic tracers reveal a peak in the 1908 layer of the Ushkovsky ice core, the occurrence of large fires in Siberia in 1908





is unlikely and suggests that the 1908 event seen in some of the Greenland ice record of ammonium originated from North America (as clearly supported by backward trajectories calculations discussed in sect. 4.1.1).

## 4.2. The last millennium

The NEEM and Summit ammonium records (annual means) spanning 1190-1990 time period are compared in
Fig. 14. As already seen for the two last centuries, the ammonium peaks tend to be better marked at NEEM than at Summit. Again, whereas a peak-to-peak comparison is rarely perfect, the two sites reveal similar events within a departure of one or two years in numerous cases. Prior 1790 and back to 1490, the agreement remains good with only a few missed events: 1719 and 1683 events recorded at NEEM but not at Summit (Fig. 14). Prior to 1490, whereas a few peaks are again not recorded in one of the two cores (e.g. 1369 and 1465 recorded at NEEM
but not at Summit, Fig. 14), a recovery of the coincidence is observed around 1200-1300 and may be due to the presence of the large sulphate peak attributed to the 1257 Samalas volcanic eruption (Lavigne et al., 2013) which served to establish the dating of the two cores.

To investigate the change in fire frequency during the last 1000 years we deployed a similar peak detection technique as in Fischer et al. (2015) based on a robust outlier detection method (Fischer et al., 1998). The only
difference was that, using singular spectrum analysis, we subtracted small background variations (on the order of 2-5 ppb) from the annual mean $NH_4^+$ data sets before peak detection (Fischer et al., 1998). For the peak detection all peaks that exceeded the median of a 200 yr window by 3.5 times the median of the absolute deviation from the median (MAD) were identified as fire peaks. In a second step the number of peaks in the 200 yr window was counted, where adjacent years identified as peaks were only counted as one. The resulting reduced peak count is
shown in Fig. 15 for the three cores.

Despite some decadal variability in each of the three records and some differences between the sites in the overall peak count level, both the two NEEM and Summit ammonium records indicate high fire activity from 1200 to 1500 and after 1850, whereas low fire activity took place particularly from 1600 to 1800 (Fig. 15). Based on the three records, this finding strengthens the previous conclusion drawn by Savarino and Legrand
(1998) on the basis of the 1190-1990 Summit record and suggesting that temporal changes coincide fairly well with the occurrence of the warm and dry climate of the Warm Medieval Period (MWP; 1200-1350) and the cold climate of the little ice age (LIA; 1600-1830). Note, that the peak count for each 200 yr window in each core should at first order follow a Poisson distribution. Hence, a peak count of about five peaks per 200 yr during the 18[th] and 17[th] century in the NEEM main core is significantly lower than its counterpart of about 12 peaks/200 yr
prior to 1400. Recently, composites series of charcoal records related to Northeast boreal fires in North America were obtained (Power et al. 2012), confirming the high fire activity at the transition from the 19[th] to 20[th] century and to a lesser extent during MWP, and the very low fire activity at the end of the LIA.

As already mentioned, it is not easy to compare these high-resolution ammonium records with low-resolution levoglucosan records. To date, the NEEM record (5-year averaged) of levoglucosan (Zennaro et al., 2014)
suggests a different picture for past changes with an outstanding maximum around 1600 (not revealed by the BC and ammonium profiles). The authors suggested that this high fire activity revealed by the levoglucosan NEEM record was caused by extensively dry conditions in Asia despite the long transport pathway to Greenland. More work is needed to elucidate why levoglucosan (and neither BC nor ammonium) would be able to record Siberian fire activity in Greenland ice.





### 4.3. Younger Dryas/Holocene fires

Several ammonium events were detected during the cold Younger Dryas (YD) climate (11.6-12.8 kyrs BP) at Summit (Fuhrer et al., 1996) and North GRIP (Fischer et al., 2015). In their study of the origin of ammonium in Greenland, Melott et al. (2010) suggested that the Haber-like production of ammonium (see Sect. 4.1.2) may have been the cause of the large ammonium peak detected at the onset of the Younger Dryas (YD) in the Summit GISP deep ice core. In their discontinuous study of the deep GRIP Summit ice core, Legrand and De Angelis (1995) measured carboxylates along four events that occurred during the YD period, including the largest one dated to 12,610 yrs BP and detected by CFA continuous sampling. As seen in Fig. 5, these four ammonium events (open triangles on the formate panel) do not differ from other events with a large increase of formate, the 12,610 yrs BP event corresponding to the highest ammonium value of 210 ppb. We therefore rule out the hypothesis of a Haber-like production of ammonium induced by a comet for the ammonium event having occurred at the onset of the YD.

In a recent study, on the basis of a low-resolution levoglucosan record obtained along the deep NEEM ice core (Table 1), Zennaro et al. (2015) reported low fire activity during the YD and the preceding period of Bolling-Allerod (BA,12.8-15 kyrs BP) followed by a steady increase during most of the Holocene (until 2.5 yrs BP) that they attributed to changes in fire emissions at the scale of the entire Northern Hemisphere. Although again it is not easy to compare this low-resolution Greenland levoglucosan record with high-resolution ammonium Greenland records, it has to be noted that the Greenland ammonium records indicate a different picture. Fischer et al. (2015) on the basis of the high-resolution (10-110 kyrs BP) ammonium record obtained along the North GRIP deep ice core indicated an increase of fire activity at the onset of BA, followed by a weak decrease during the YD, and a second increase toward the beginning of the Holocene. For the Holocene, on the basis of the ammonium record at Summit, Fuhrer et al. (1996) reported strong fire activity at 10 kyrs BP and an overall decreasing trend throughout the Holocene (Fuhrer et al., 1996). Such a decreasing trend change during the Holocene is consistent with charcoal records from the Eastern part of North America (Carcaillet et al., 2002; Marlon et al., 2013). More work is needed to elucidate why levoglucosan and ammonium suggest such different temporal trends. At this stage, we can only emphasize that the levoglucosan records seem to mirror changes of fire activity at a larger scale (Eurasia plus Canada as discussed in Sect. 4.2 for the last millennium, or even larger as discussed above for the last 15 krs) than ammonium records. If confirmed, this conclusion raises the question of the atmospheric lifetime of levoglucosan with respect to other aerosol such as BC or ammonium formate.

### 5. What we have learned: Recommendations for further investigations.

This comparison of various chemical Greenland ice core records permitted evaluation of the quality of different proxies to trace boreal fires. Ammonium, organic fractions (black carbon as well as organic carbon), and organic compounds like formate and vanillic acid are found to be good proxies whereas the use of potassium, nitrate, and carboxylates (except formate) is complicated by either post-depositional effects or existence of large non-biomass burning sources. This comparison also highlighted important factors that modulate the records of such sporadic events such as the occurrence of snowfall, surface snow wind erosion and transport between the source region of Canada and the different Greenland drill sites. From that, we make the recommendation to replicate





high-resolution CFA measurements, in priority for ammonium, TOC, and BC along ice cores extracted at different sites to provide a composite Greenland ice record.

Further work dedicated to high-resolution measurements of levoglucosan would be welcome. These measurements will enable improved understanding of the cause of the observed difference in past fire activity changes derived from levoglucosan and ammonium records. Concerning ammonium ice records, they consistently indicate Canada as the main source for fire plumes reaching Greenland and show agreement with charcoal records past fire activity changes in response to climatic fluctuations.

Finally field studies of post depositional effect would be interesting for several organic species (or groups of species).

## 6. Data availability

Data from NEEM-2011-S1, SUMMIT 2010, North GRIP 2, D4, and Humboldt can be made available for scientific purposes upon request to J. McConnell at the Desert Research Institute. CFA analyses of ammonium and formaldehyde at GRIP as well as CFA ammonium on the NEEM deep ice core can be made available for scientific purposes upon request to H. Fischer at the Physics Institute and Oeschger Center for Climate Change Research at Bern. Ion chromatography data from GRIP, GRIP 1993, and Eurocore can be made available upon request to M. Legrand at the Laboratory of Glaciologie et Géophysique de l'Environnement at Grenoble. Canadian fire data are made available by the Canadian Forest Service under http://cwfis.cfs.nrcan.gc.ca/datamart.

**Aknowledgements:**

We would like to thank Katrin Fuhrer who did CFA chemical measurements at GRIP, Martine De Angelis who helped for IC measurements at GRIP and EUROCORE. Collection, analyses (or reanalysis), and interpretation of the Humboldt, D4, Summit 2010, NEEM 2011 S1, and Tunu 2013 ice cores were funded by U.S. NSF grants 0856845, 0909541, 1023672, and 1204176. Analyses and interpretation of the NGRIP2 core was support by the University of Oxford's John Fell Fund. We gratefully acknowledge the contributions of the many drilling and logistics personnel involved in the collection and processing of these cores, as well as the students and staff of Ultra-Trace Chemistry Laboratory at the Desert Research Institute. The long-term financial support of ice core research at the University of Bern by the Swiss National Science Foundation (SNF) is gratefully acknowledged. NEEM is directed and organized by the Centre of Ice and Climate at the Niels Bohr Institute and US NSF, Office of Polar Programs. This research also is supported by funding agencies and institutions in Belgium (FNRS-CFB and FWO), Canada (NRCan/GSC), China (CAS), Denmark (FIST), France (IPEV, CNRS/INSU, CEA and ANR), Germany (AWI), Iceland (RannIs), Japan (NIPR), South Korea (KOPRI), The Netherlands (NWO/ ALW), Sweden (VR), Switzerland (SNF), the United Kingdom (NERC) and the USA (US NSF, Office of Polar Programs). We would like to thank Anne-Laure Daniau (EPOC laboratory in Bordeaux, France) for very helpful discussions on charcoal records. She also organized the special session SGF/AFEQ-CNF INQUA on Feux de végétation, poussières minérales et changement climatique: observation et modélisation (2013, Paris, France, coordinated by A-L Daniau and Y. Balkanski) during which were launched discussions on fire proxies in ice. Lastly, we thank John Little (Natural Resources Canada) for his assistance with the Canadian National Fire



Database (CNFD).

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



**Table 1.** Various Greenland ice records that were used to investigate past change of forest fires. In bold are species chosen by the authors of these previous studies as good fire proxies in ice (others were found to be influenced by fires but not enough to be considered as good proxies; see discussion in Sect. 3.2).

| Sites (snow accumulation) | Locations (elevation) | Time-period | Resolution | Proxies | References |
|---|---|---|---|---|---|
| Summit (GISP2) (~22 g cm⁻² yr⁻¹) | Central Greenland (3240 m asl) | 1750-1980 | Mostly biannual (Subsampling) | **$NH_4^+$** $K^+$, $NO_3^-$ | Whitlow et al. (1994) |
| Summit (GRIP 93) (~22 g cm⁻² yr⁻¹) | Central Greenland (3240 m asl) | 1767-1993 | Seasonally resolved (Subsampling) | **$NH_4^+$ and $HCOO^-$** $K^+$, $CH_3COO^-$, $C_2O_4^{2-}$ | Legrand and De Angelis (1996) |
| Summit (Eurocore) (~22 g cm⁻² yr⁻¹) | Central Greenland (3240 m asl) | 1193-1767 | Seasonally resolved (Subsampling) | **$NH_4^+$ and $HCOO^-$** $C_2O_4^{2-}$, $NO_3^-$ | Savarino and Legrand (1998) |
| Summit (GRIP) (~22 g cm⁻² yr⁻¹) | Central Greenland (3240 m asl) | 0-3 kyr BP, 8-12 kyr BP | Seasonally resolved (CFA) | **$NH_4^+$** | Fuhrer et al. (1996) |
| North GRIP (~19 g cm⁻² yr⁻¹) | Central Greenland (2917 m asl) | 10-110 kyr BP | Yearly averaged CFA values | **$NH_4^+$** | Fischer et al. (2015) |
| D20 (~41 g cm⁻² yr⁻¹) | SE Greenland (2625 m asl) | 1750-1980 | Seasonally resolved (Subsampling) | **$NH_4^+$** $K^+$, $NO_3^-$ | Whitlow et al. (1994) |
| D4 (~41 g cm⁻² yr⁻¹) | W central Greenland (1985 m asl) | 1788-2002 | Monthly (BC) (CFA) Annual (VA) (CFA) | **Black carbon (BC) and vanillic acid (VA)** | McConnell et al. (2007) |
| NEEM (deep core, 2011-S1) (~20 g cm⁻² yr⁻¹) | NW Greenland (2454 m asl) | 0-2 kyr BP | 5 years Monthly (BC) (CFA) | **Levoglucosan, $NH_4^+$** BC | Zennaro et al. (2014) |
| NEEM (deep core) (~20 g cm⁻² yr⁻¹) | NW Greenland (2454 m asl) | 0-15 kyr BP | 5-50 years | **Levoglucosan** | Zennaro et al. (2015) |





**Table 2.** Averaged fraction of time spent by air masses over different forested areas as calculated for June and July backward trajectories (five and ten days) arriving 500 m above Summit and NEEM sites. Calculations were made for 25 years between 1948 and 1991 (1948–1953, 1959–1965, 1978–1983, and 1986–1991) and during a few years characterised by large fires in Canada (1961, 1979-1981, and 1989: Fig. 10).

| Site (Years) | Western Canada | Eastern Canada | Canada | Western Europe | Siberia |
|---|---|---|---|---|---|
| Summit (25 years) | 0.6% (5 days) 3.0% (10 days) | 2.6% (5 days) 4.6% (10 days) | 3.2% (5 days) 7.6% (10 days) | 0.3% (5 days) 1% (10 days) | - 0.6% (10 days) |
| NEEM (25 years) | 0.3% (5 days) 1.6% (10 days) | 1.0% (5 days) 2.6% (10 days) | 1.2% (5 days) 4.2% (10 days) | - 0.4% (10 days) | - 0.8% (10 days) |
| Summit (Fires) | 0.6% (5 days) 3.2% (10 days) | 2.6% (5 days) 4.9% (10 days) | 3.3% (5 days) 8.2% (10 days) | 0.1% (5 days) 1% (10 days) | - 0.6% (10 days) |
| NEEM (Fires) | 0.1% (5 days) 1.5% (10 days) | 1.2% (5 days) 2.5% (10 days) | 1.3% (5 days) 4.0% (10 days) | - 0.4% (10 days) | - 0.5% (10 days) |



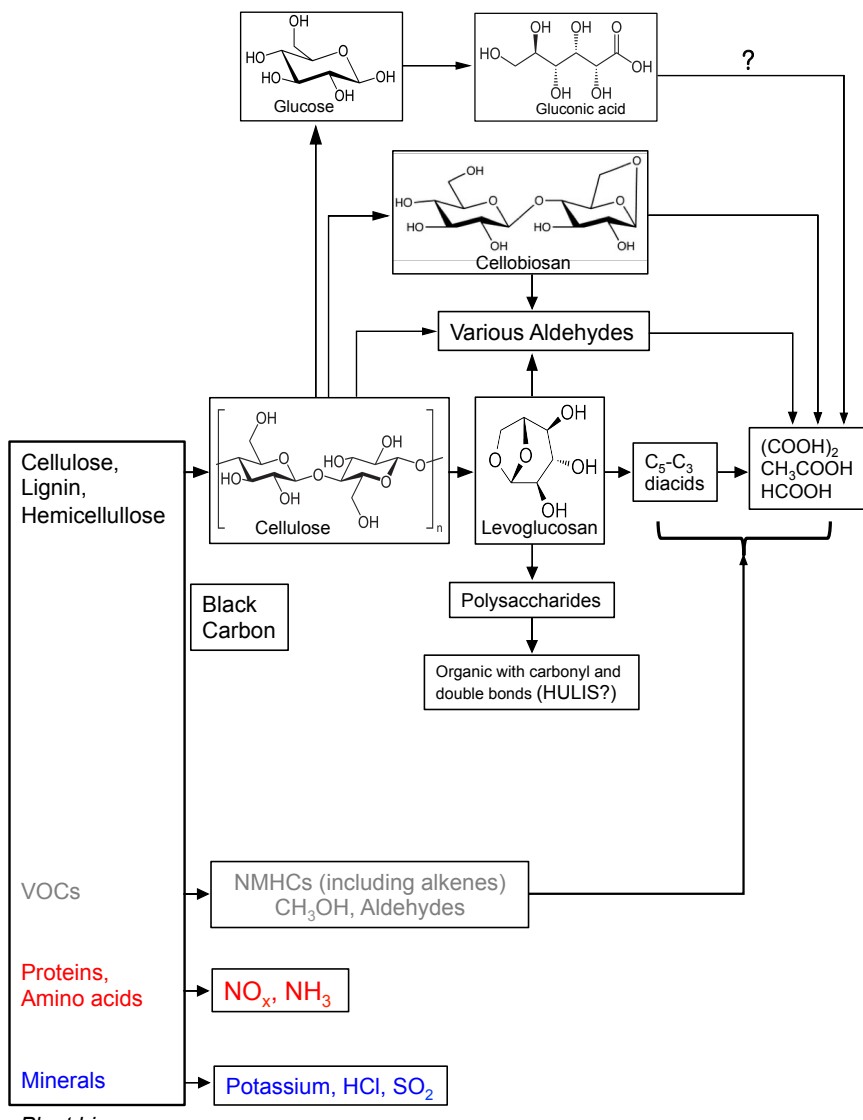

**Figure 1:** Proposed chemical pathways for formation of various chemical species identified in smoke aerosols.

Adapted from Gao et al. (2003) and references therein, Kawamoto et al. (2003) and Abella et al. (2007).





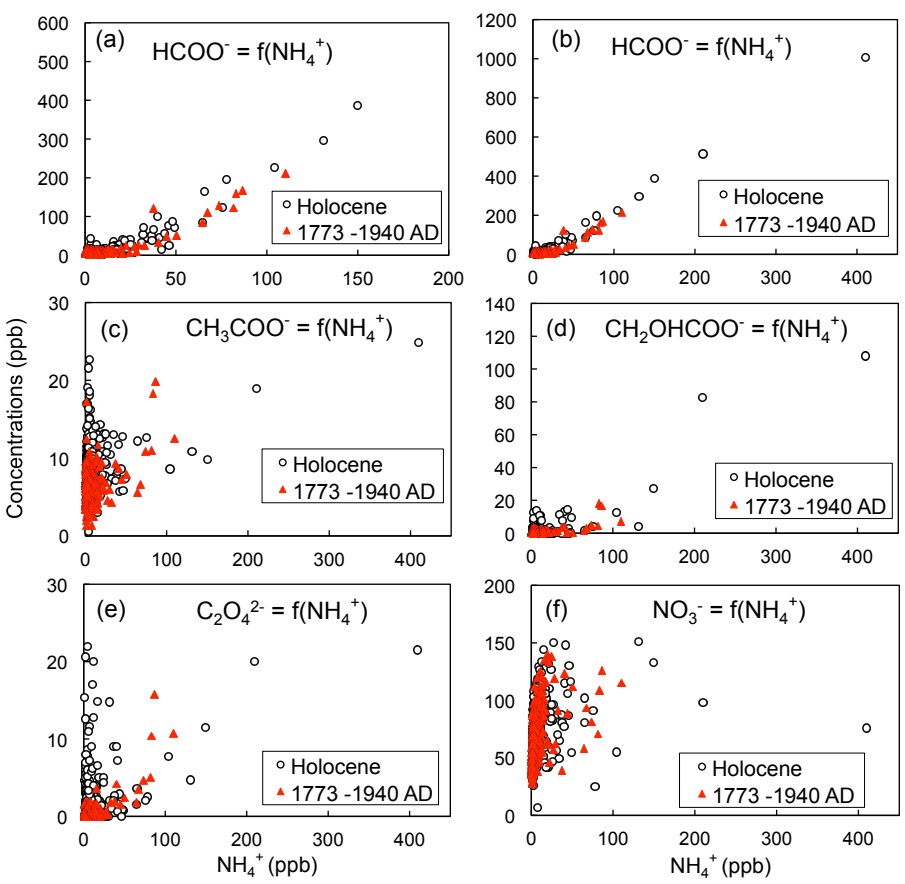

**Figure 2:** Concentrations of various carboxylates (a to e) and nitrate ($NO_3^-$, f) as a function of ammonium ($NH_4^+$) in Summit snow and ice layers. $HCOO^-$ is formate, $CH_3COO^-$ is acetate, $CH_2OHCOO^-$ is glycolate, and $C_2O_4^{2-}$ is oxalate. For formate, two panels (a and b) are shown to detail the relationship at ammonium concentrations lower than 200 ppb (panel a). The red triangles correspond to the layers deposited between 1773 and 1940 AD (520 values along the GRIP 93 firn core: Table 1); the black circles correspond to the Holocene period along the GRIP deep ice core (660 values between and 355 and 11,600 years BP).





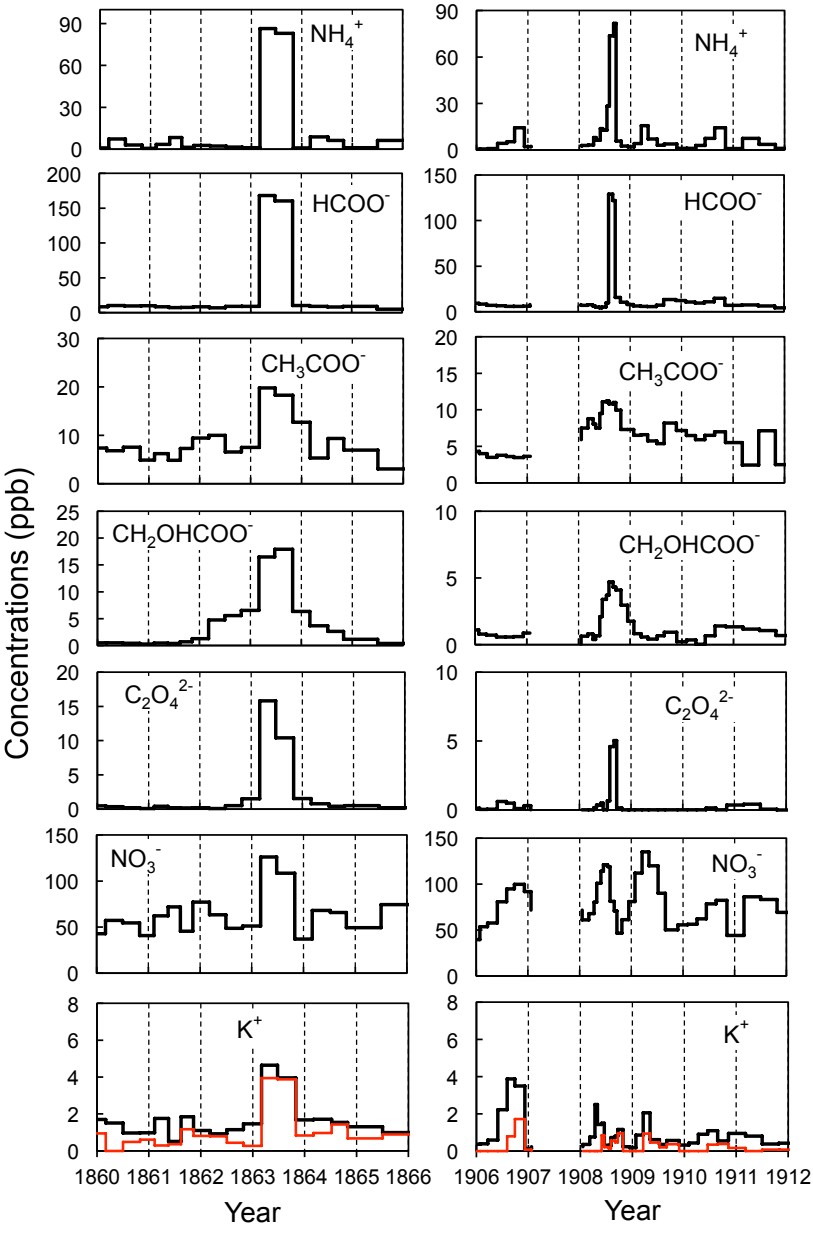

**Figure 3:** Chemical fingerprint of two ammonium events dated in the GRIP 93 firn core at 1863 AD (left) and 1908 AD (right). $NH_4^+$ is ammonium, $HCOO^-$ is formate, $CH_3COO^-$ is acetate, $CH_2OHCOO^-$ is glycolate, $C_2O_4^{2-}$ is oxalate, $NO_3^-$ is nitrate, and $K^+$ is soluble potassium. For soluble potassium, the black and red curves refer to total and fine fraction, respectively (Sect. 3.2.3)





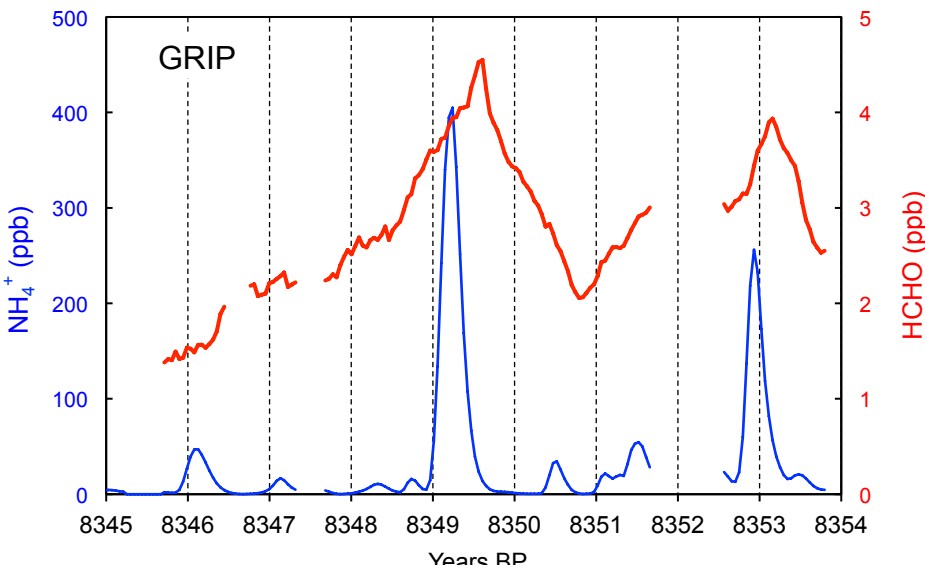

**Figure 4:** An example of formaldehyde (HCHO) perturbations detected at Summit (GRIP deep ice core) along two ammonium events.





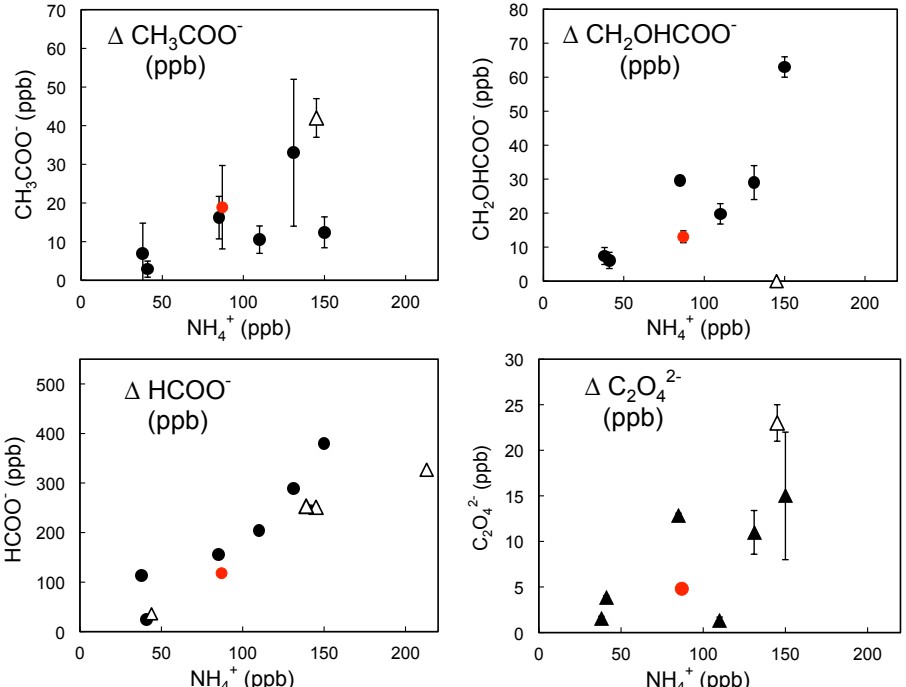

**Figure 5:** Enhancement of carboxylates above their background levels as a function of ammonium along several fire events detected in snow and ice layers deposited at Summit (GRIP 93 firn core and the deep GRIP ice core). HCOO⁻ is formate, CH₃COO⁻ is acetate, CH₂OHCOO⁻ is glycolate, and C₂O₄²⁻ is oxalate. The black circles refer to events that occurred during the Holocene (0 to 11,400 yrs BP), the red dots refer to the 1908 AD event shown in Fig. 3. Open triangles correspond to four ammonium events that occurred during the Younger Dryas (11,600 to 12,600 yrs BP). For acetate and glycolate, calculation of enhancements considered the smoothing of the perturbation around the ammonium perturbation. Vertical bars are uncertainties in calculating the enhancements above background values.





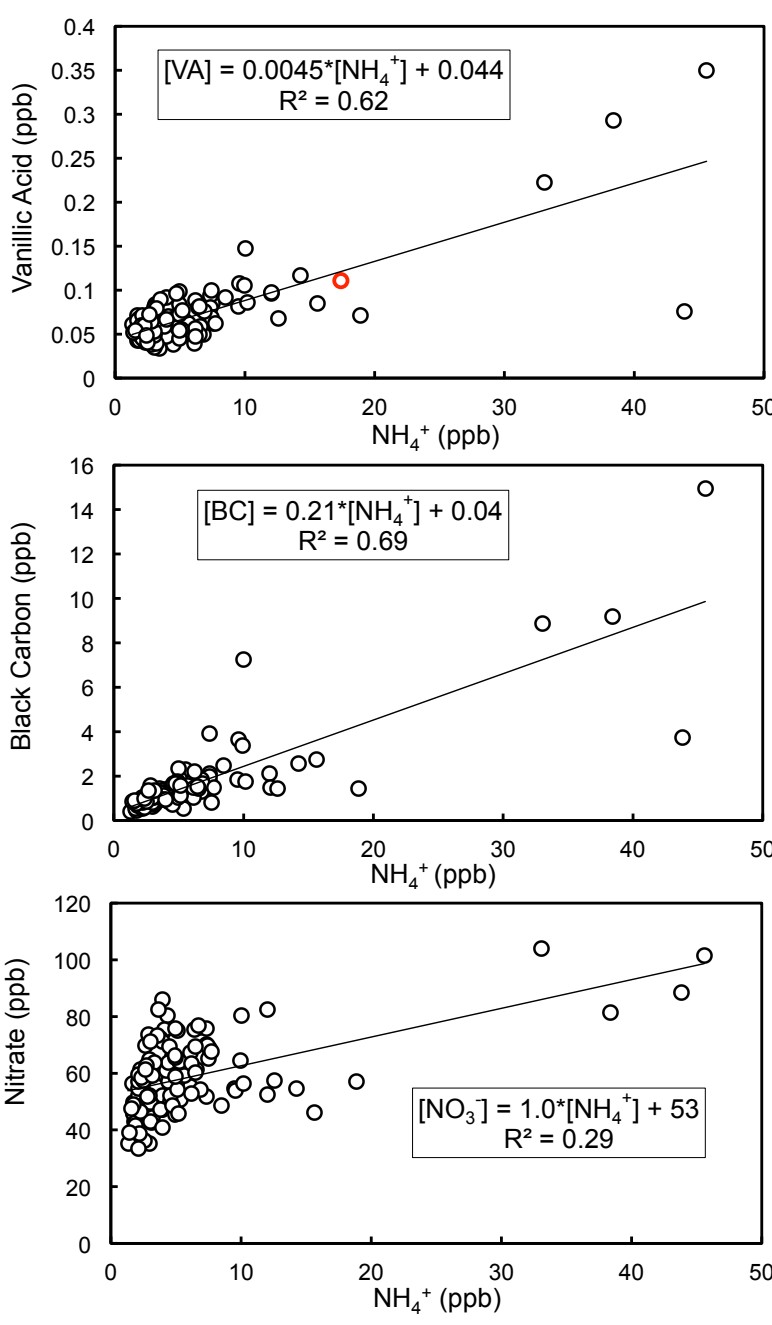

**Figure 6:** Annual mean concentrations of vanillic acid, black carbon, and nitrate versus ammonium in snow layers deposited at D4 during pre-industrial times (from 1740 to 1870). For vanillic acid, we also report the point corresponding to the 1908 event (red circle).





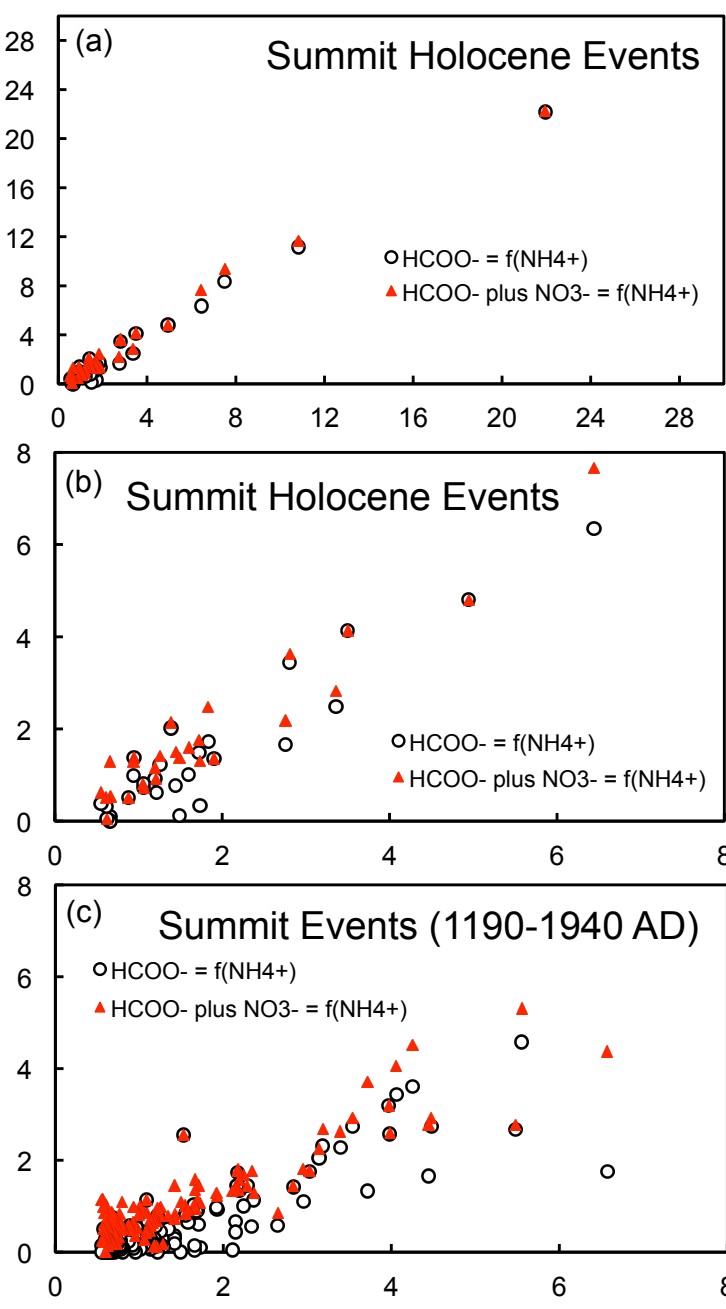

**Figure 7:** Ammonium versus formate (formate plus nitrate) along $NH_4^+$ events at Summit. All concentrations are expressed in $\mu Eq\ L^{-1}$.



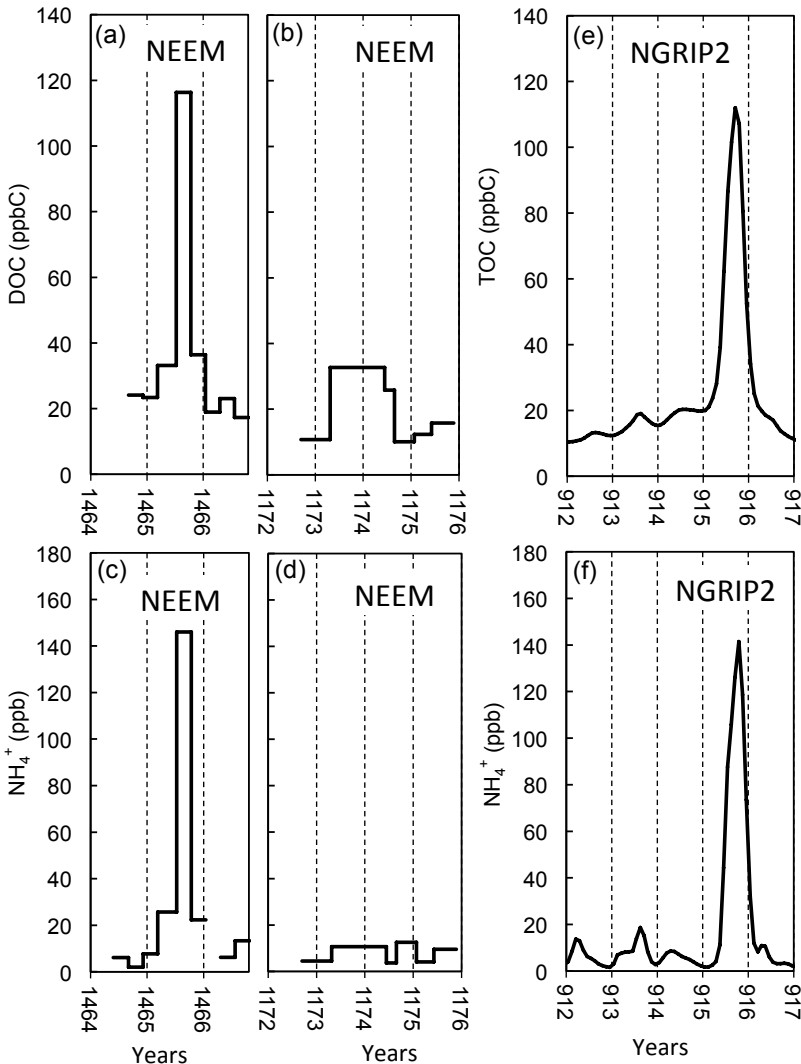

**Figure 8:** Example of organic carbon (OC) changes along fire events in Greenland ice. (a) to (d) are from the NEEM-2011-S1 ice core for which OC was measured with a UV oxidation method (Sect. 2) and ammonium on discrete samples analysed with ion chromatography. (e) and (f) are from NGRIP 2 ice for which ammonium was measured with a CFA system and OC was investigated with a CFA coupled to a total organic carbon analyser (Sect. 2).





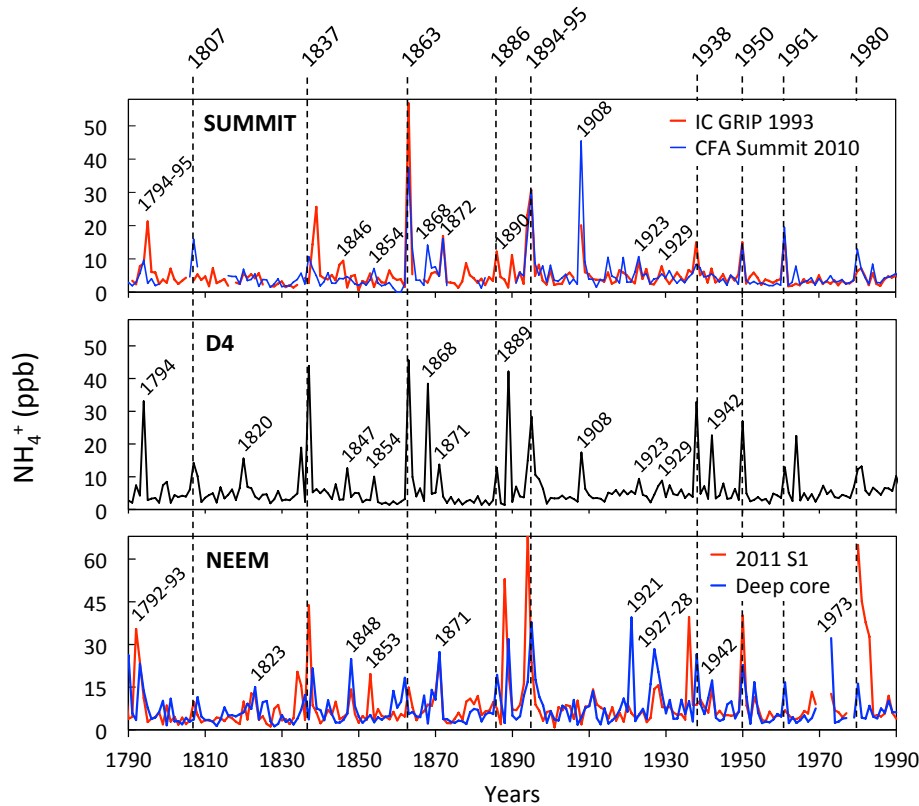

**Figure 9:** Comparison of several ammonium records obtained at Summit (IC for GRIP 1993, CFA for Summit 2010), D4 (CFA) and NEEM (CFA). All records are seasonally resolved, but data are presented here as annual means. Dashed lines denote the years for which a peak is identified at all sites. The IC GRIP 93 ammonium data are from Legrand and De Angelis (1996).



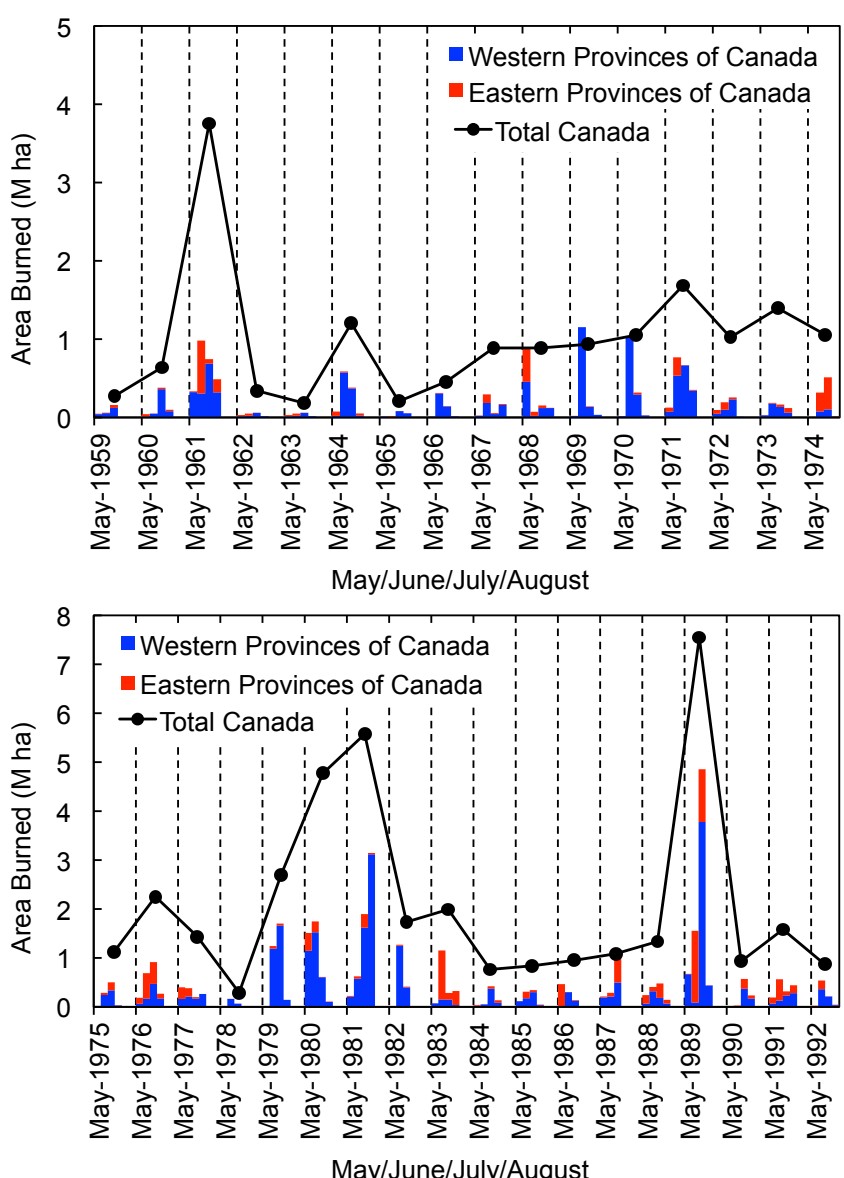

**Figure 10:** Area burned in Canada from 1959 to 1992 in summer (from May to August), distinguishing western (in red) and eastern (in blue) provinces (data from CNFD: Sect. 4.1.1). The black curve refer to total area burned in Canada (adapted from Van Wagner, 1988).



**Summer 1978, Backward trajectories**

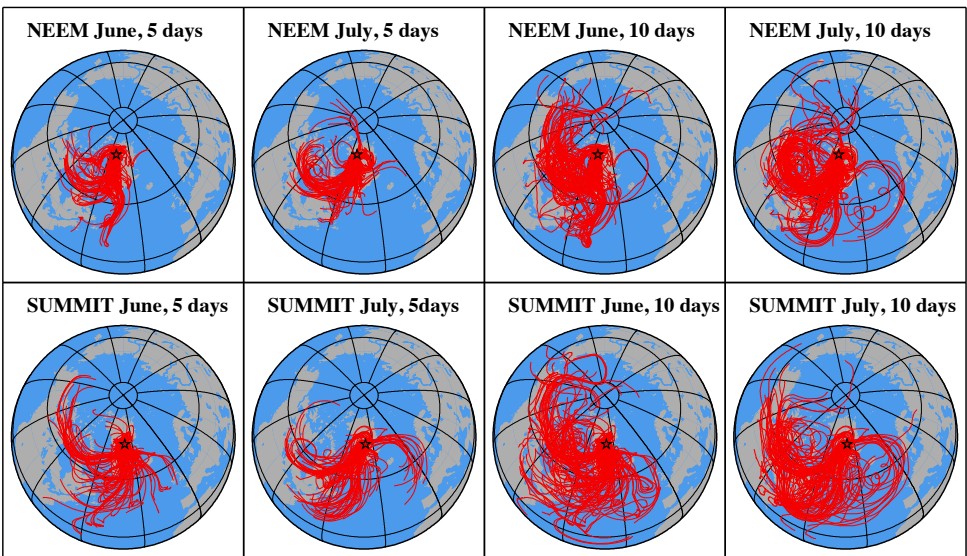

**Figure 11:** Five- and ten-day backward air mass trajectories arriving at the NEEM and Summit sites in June and July 1978 (site arrival at 500 m above ground level).





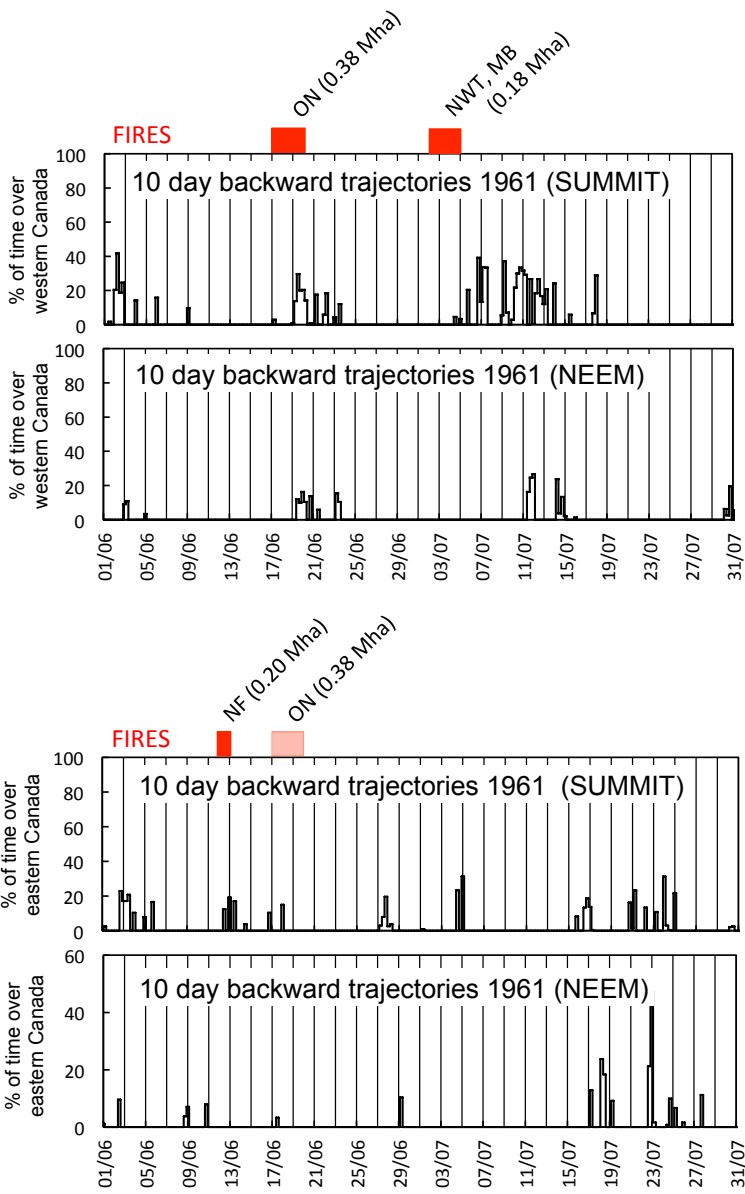

**Figure 12:** Fraction of time spent by air masses arriving at Summit and NEEM (ten-day backward trajectories) at 500 m agl and having traveled over a forested area of western (top) and eastern (bottom) provinces of Canada during the severe fire conditions of June/July 1961 (Fig. 10). Some of the largest fires are indicated on top of the panels (red symbols) with corresponding burned area: Ontario and Northwest Territories in the West, Newfoundland and Labrador in the East. The Ontario fire was located in the western portion of the province and we report the red symbol on both panels.





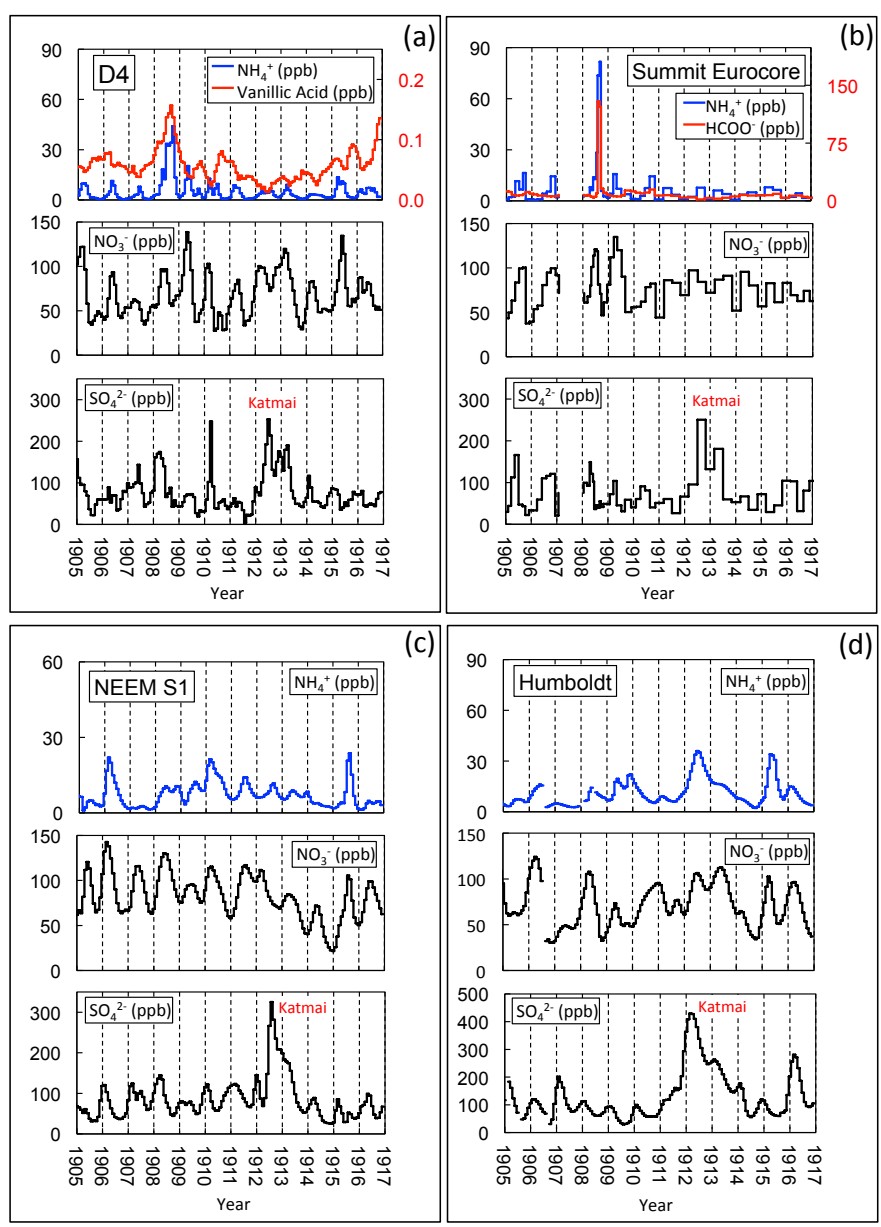

**Figure 13:** High-resolution 1905-1916 records of ammonium, nitrate, and sulfate at D4 (a), Summit (GRIP 93) (b), NEEM 2011 S1 (c), and Humbold (d). At Summit and D4, data on carboxylic acids also were available, formate at Summit (Legrand and De Angelis, 1998), and vanillic acid at D4 (McConnell et al., 2007). Note the presence of the Katmai volcanic layer (1912) in the sulfate records, permitting accuracy of the dating to be <1 yr during this sequence of years.



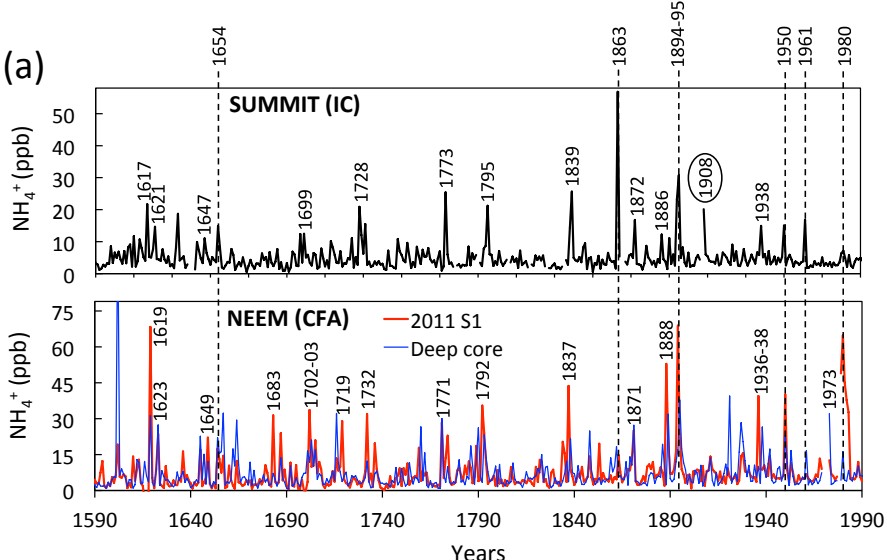

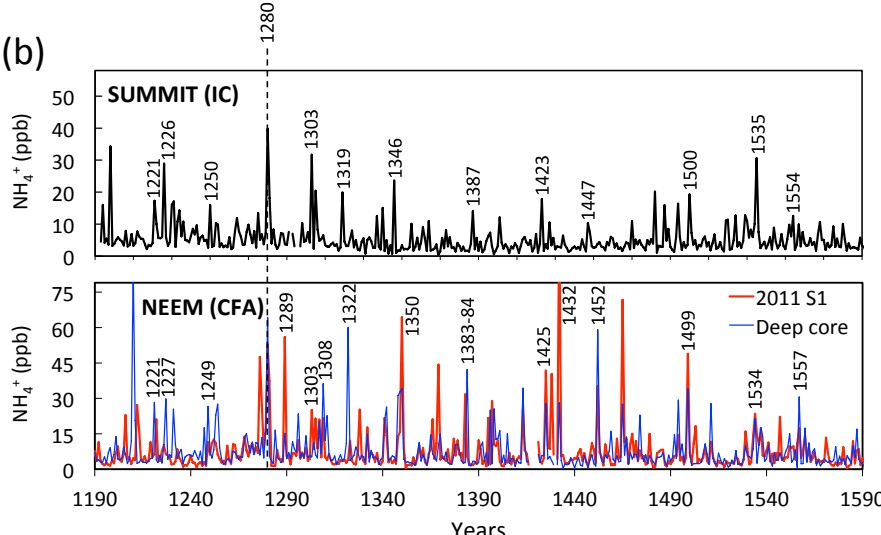

Figure 14: Comparison of ammonium records obtained at Summit and NEEM (NEEM-2011-S1 and the deep core) from 1590 to 1990 (a) and from 1190 to 1590 (b). All records are seasonally resolved, but data are presented here as annual means. Dashed lines denote the years for which a peak is identified in both cores. The Summit data are from Legrand and De Angelis (1996) and Savarino and Legrand (1998).





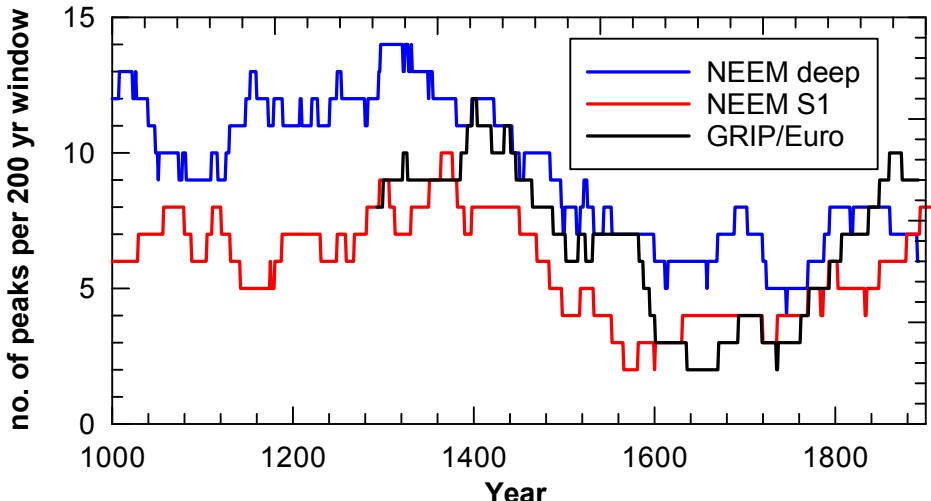

**Figure 15:** Fire peak frequency in a running 200-yr window for the Eurocore at Summit (black), the NEEM deep core (blue) and the NEEM 2011 S1 core (red). Peaks detected in adjacent years were only counted as one.

