# Peer review of "Boreal fire records in Northern Hemisphere ice cores: A review"

_Climate of the Past, 2016_

## Referee Comment (RC1) · P. Vallelonga (Referee) · 12 Aug 2016

General comments

The manuscript "Boreal fire records in Northern Hemisphere ice cores: A review" by Legrand and co-authors provides a thorough evaluation of the major proxies used to evaluate biomass burning activity from Greenland and alpine ice cores. There is an appropriate treatment of the chemical precursors to biomass burning proxies and the fire processes involved in their emission. The manuscript provides a good summary of the state-of-the-art for these techniques as well as an outlook on problems and opportunities for future research. The manuscript could benefit from some small improvements suggested below in the specific comments, but is already in a very good state. I will be happy to provide NEGIS data to develop the discussion regarding the 1908 event

described below.

Specific comments

Continuous measurement of TOC. The Sievers 900 technique described in section 2 is novel and should be described in additional detail. Given the technique is susceptible to contamination by drill liquid, it would be good to show data from the dry-drilled TUNU core, for which ammonium data is also available.

Data used for evaluation. The treatment of geographical distribution of ammonium signals in Greenland (discussed in section 4 and shown in figures 9, 14, 15) would benefit from the inclusion of ice core data from Northeast Greenland, such as the TUNU and NEGIS (Vallelonga et al., 2014, The Cryosphere, doi: 10.5194/tc-8-1275-2014) sites. For example, the 1908 ammonium peak is visible in both the TUNU and NEGIS records, pointing to a widespread signal across Greenland, that is curiously absent at NEEM. The absence of such a signal at NEEM may be indicative of an atmospheric transport path that does not arrive at NEEM and is perhaps distinct from that which transports aerosols from North America.

Levoglucosan (section 3.2.5) The authors have rightly pointed out that levoglucosan records are not available in annual or sub annual resolution, which limits comparisons to other biomass burning proxies such as ammonium or BC. Given that levoglucosan measurements are relatively slow and laborious, it is unlikely that annually-resolved data will be produced in the foreseeable future. Consequently, it is important for a critical review such as this one to provide a critical evaluation of the available data. The authors should be more explicit regarding their observation that levoglucosan data produced to date is inconsistent with ammonium and BC records (sections 4.2 and 4.3) and this should be highlighted as an important concern to be resolved in future studies.

---

## Referee Comment (RC2) · Anonymous Referee #2 · 19 Aug 2016

The authors aim to determine if fire records vary spatially between northern versus central Greenland sites as well as if transport patterns of the various biomass burning markers affected the fire records. Such a comparison of different fire markers in Greenland ice cores is a worthwhile goal, although the authors should highlight that some of their assumptions and conclusions are specific to Greenland. The first two-thirds of the paper are detailed and provide well-rounded explanations, while the final third feels rushed with a conclusion of the necessity of extracting more ice cores across Greenland for better spatial representation of these markers.

The current studies of biomass burning in ice cores are exponentially increasing. I applaud the authors for systematically investigating the differences between these proxies in a specific region as such comparisons help the ice core and fire science communities as a whole. This paper adds a beneficial review to the literature.

General comments:

The authors have extensive knowledge and a corresponding publication record of investigating ammonium in Greenland ice cores. The authors therefore strongly emphasize ammonium throughout this investigation of biomass burning proxies in Greenland ice cores and compare every other chemical fire marker to ammonium (e.g. Figure 2 and all sections of 3.2). However, the authors should justify why they use ammonium as the marker against which they compare every other record. The paper circuitously mentions (but not until section 3.3) that ammonium is mainly a viable fire marker in the poles as these regions are located far enough away from biogenic sources of ammonium, but this important point should be mentioned very early in the introduction. The authors could strengthen the entire paper if they are able to demonstrate that biogenic ammonium does not influence the ammonium record in Greenland.

Although the authors provide an overview on the known atmospheric lifetime of many of the species, the authors assume that each proxy will follow the same back trajectory. Some markers (ie vanillic acid) may be able to be injected higher into the atmosphere than heavier particles such as BC. These different injection heights can influence their transport and thereby also influence their atmospheric lifetime. The authors mention the difference in plume altitude with respect to the relative abundance of NO compared to NH3, but (page 8) but do not mention this important parameter when investigating the contributions of the other biomass burning markers. A discussion on injection heights and depositional differences between markers should be included in the paper.

Section 4.1.2 distracts the reader from the central aims of the paper as the level of detail included in this section is much greater than that provided in other sections, and as the evidence for or against the Tunguska asteroid impact is only obliquely related to biomass burning. Significantly shortening or even omitting this section will help "tighten" the manuscript.

Specific comments:

Page 2, Lines 4 and 5: What spatial and temporal differences occurred since 1900?

Page 2, Lines 9 and 10: The first known human-origin fires occurred 1 million years ago (Berna et al., PNAS, 2012). Although he scientific community is actively seeking 1 million year old ice, currently the record of human-origin fires is longer than any ice core record. Please revise or omit this statement.

Page 2, Lines 20-23: Mentioning that paleo-fire records are available for these distinct regions of North America is somewhat misleading. Using the publicly-available R paleofire package (Blarquez et al., Computers & Geosciences, 2014), scientists are able to compile charcoal data into any regions of their choosing.

Page 3, Lines 15-17: Do you mean that high-resolution records are limited form the present day until the last millennium OR that these high resolution records only exist for the past millennium?

Page 5, Lines 37-29: Please state why it is important that nitrogen emissions are dominated by ammonia and not NOx.

Page 6, Line 6-10: Do you mean that burning is not expected to produce levoglucosan in flaming conditions regardless of the vegetation type?

Page 8, Line 1-6. If the R2 of formate versus ammonium is higher than the R2 of vanillic acid versus ammonium, why then use vanillic acid? Do you prefer vanillic acid because formic acid is produced during smoke plume aging (page 7)? Why do you not show formate in Figure 6?

Section 3.2.5. The authors state studies that are towards the low end of the spectrum of atmospheric lifetimes for levoglucosan. Slade and Knopf, 2013 mention that under atmospheric background conditions, levoglucosan likely has an atmospheric lifetime of 2 weeks, while Bai et al., 2013 suggest that levoglucosan may have a mean atmospheric lifetime of 26 days.

Section 3.2.6: Why do you designate these different groups if you do not use them

later in the paper? Mentioning that certain markers are "grouped" or "similar" rather then designating specific groups (ie Group 1) may help the reader.

Section 3.3: Why do you not include the Akedemii Nauk (Siberia) ice core? This core may provide a link between the Greenland cores and the more temperate Siberian and Alaskan core that you include in the paper.

Page 13: If there is little difference between the 0, 250 and 500 m above ground level back trajectories, why do you choose the 500 m trajectories? You are investigating material that is deposited at the surface (0 m above ground level).

Figures 11 and 12: Although you mention the source of your data in the paper, please also mention this data source in the figure captions.

Page 16 Lines 34-39: The authors spent multiple pages in the beginning of the paper outlining differences between what individual proxies record (ie BC results from flaming fires and/or fossil fuel burning while levoglucosan is produced from smoldering fires and where ammonium contains a biogenic emissions source). I agree that "more work is needed to elucidate why levoglucosan (and neither BC nor ammonium) would be able to record Siberian fire activity in Greenland ice " is necessary, but the authors themselves provide plausible explanations earlier in the paper. In Section 4.3 the authors mention that levoglucosan may "mirror changes of fire activity at a larger scale. . .. than ammonium records" which is another explanation for these differences. The authors should include these possibilities during the discussion in Section 4.2 (lines 34-39).

I completely understand the difficulties in writing an academic manuscript in another language. However, in many locations in the paper the writing style gets in the way of understanding the science (e.g. page 5). Multiple co-authors are native English speakers and I strongly urge these co-authors to carefully edit the paper to remove instances of passive voice, flipped clauses and noun-adjective pair placements.

Technical corrections:

Page 3, Line 3: Place "biomass burning source" or some equivalent after "this".

Page 3, Line 39: Place "comparison" after "this".

Sometimes temperatures are recorded in degrees C and sometimes in degrees K. Please be consistent throughout the paper.

Page 8, Line 17: Replace "unit" with "unity"
* * *

---

## Author Comment (AC1) · 21 Sep 2016

**Response to P. Vallelonga (Referee #1)**

General comments

The manuscript "Boreal fire records in Northern Hemisphere ice cores: A review" by Legrand and co-authors provides a thorough evaluation of the major proxies used to evaluate biomass burning activity from Greenland and alpine ice cores. There is an appropriate treatment of the chemical precursors to biomass burning proxies and the fire processes involved in their emission. The manuscript provides a good summary of the state-of-the-art for these techniques as well as an outlook on problems and opportunities for future research. The manuscript could benefit from some small improvements suggested below in the specific comments, but is already in a very good state. I will be happy to provide NEGIS data to develop the discussion regarding the 1908 event described below.

Specific comments

Continuous measurement of TOC. The Sievers 900 technique described in section 2 is novel and should be described in additional detail. Given the technique is susceptible to contamination by drill liquid, it would be good to show data from the dry-drilled TUNU core, for which ammonium data is also available. *We agree and in the revised version we added (section 2) "Recent improvements in sample handling and other techniques in the CFA system at DRI enable arguably the first reliable high-resolution measurements of total organic carbon (TOC). In this new method, a Sievers 900 TOC analyzer is coupled to an ice core melter. The constantly flowing sample stream (isolated from any interaction with laboratory air and after minimal contact with plastics and other sources of contamination in the flow lines) is analysed within a few minutes of initial melting. Total OC is determined in the Sievers 900 analyzer as the difference in total carbon and inorganic carbon. UV radiation and ammonium persulfate are used to oxidize organic compounds to $CO_2$, and the $CO_2$ is measured with a patented selective membrane-based conductometric detection method (details available in the Sievers 900 Series Total Organic Carbon Analyzers, Operation and Maintenance Manual, GE Analytical Instruments, 2011). We emphasize that for both DOC (Legrand et al., 2013) and TOC, measurements are most reliable in ice cores drilled without use of organic drilling fluids and in samples from below the pore close-off depth where potential contamination from circulating modern air through the core is eliminated."*

*Concerning the possible contamination when a drill fluid is used we wrote (section 3.2.4) : "As seen in Fig. 8, a very similar picture was obtained at DRI using CFA coupled with a TOC analyser. Although limited, these data highlight three important points. First, the similar OC background levels observed in the dry-drilled Summit-2015-Place ice core and in the wet-drilled North GRIP2 and NEEM cores (Fig. 8), suggests that the use of drill fluid has not significantly contaminated the North GRIP2 and NEEM ice."*

Data used for evaluation. The treatment of geographical distribution of ammonium signals in Greenland (discussed in section 4 and shown in figures 9, 14, 15) would benefit from the inclusion of ice core data from Northeast Greenland, such as the TUNU and NEGIS (Vallelonga et al., 2014, The Cryosphere, doi: 10.5194/tc-8-1275- 2014) sites. For example, the 1908 ammonium peak is visible in both the TUNU and NEGIS records, pointing to a widespread signal across Greenland, that is curiously absent at NEEM. The absence of such a signal at NEEM may be indicative of an atmospheric transport path that does not arrive at NEEM and is perhaps distinct from that which transports aerosols from North America. *This*

*point was further discussed following email exchanges between Paul and Michel. Since the recommendation of the second reviewer was to minimize the discussion of the Tunguska and that, at the opposite of the case of Summit, NEEM and D4, for which continuous records covering the last 100 years or the last millennium are already published and presented in this paper (see Figures 9 and 14), we decided that we will mention at the end of the first paragraph of section 2 (Previously published and unpublished data) as follows: "Finally, recent and new chemical investigations, including ammonium (not shown), document biomass-burning fallout in ice cores extracted at the NEGIS (Vallelonga et al., 2014) and Tunu-2013 sites (Sigl et al., 2015). Ongoing studies conducted on northeast Greenland ice cores will complement results discussed here."*

Levoglucosan (section 3.2.5) The authors have rightly pointed out that levoglucosan records are not available in annual or sub annual resolution, which limits comparisons to other biomass burning proxies such as ammonium or BC. Given that levoglucosan measurements are relatively slow and laborious, it is unlikely that annually-resolved data will be produced in the foreseeable future. Consequently, it is important for a critical review such as this one to provide a critical evaluation of the available data. The authors should be more explicit regarding their observation that levoglucosan data produced to date is inconsistent with ammonium and BC records (sections 4.2 and 4.3) and this should be highlighted as an important concern to be resolved in future studies. *The discussion on Levoglucosan was updated (also considering comments from the second reviewers). First, at the end of section 3.2.5 (Levoglucosan) we wrote: "These laboratory studies suggest an atmospheric lifetime of levoglucosan against chemical degradation of approximately two days (Lai et al., 2014), or from two days to two weeks (Slade and Knopf, 1013). Given the lower range of estimated levoglucosan lifetime against chemical degradation we cannot rule out that chemical loss represent a significant loss for levoglucosan additional to the depositional loss that would apply to all biomass burning aerosol. "*

*Second, at the end of paragraph 4.2, we added "Note that this conclusion conflicts with the hypothesis of a significant chemical degradation of levoglucosan suggested by laboratory studies (Sect. 3.2.5). More work is needed to elucidate why (1) levoglucosan might reflect Siberian fire activity in Greenland ice but not BC or ammonium, and (2) the finding that the levoglucosan record seems to mirror changes of fire activity at a larger spatial scales (Eurasia plus Canada) than ammonium records."*

*Third, in the conclusion we added "Further work dedicated to high-resolution measurements of levoglucosan also would be welcome. Such measurements would enable improved understanding of the cause of the observed difference in past fire activity changes derived from levoglucosan and ammonium records. A large amount of ice would be needed to achieve high-resolution levoglucosan ice core profiles, but sampling Greenland snow in snow-pits would be a useful alternative. For instance, a 5 m depth pit at Summit would span the last 10 years during which numerous fire events have occurred in North America and Siberia that are well documented by satellite observations. "*

---

## Author Comment (AC2) · 21 Sep 2016

**Response to Anonymous Referee #2**

The authors aim to determine if fire records vary spatially between northern versus central Greenland sites as well as if transport patterns of the various biomass burning markers affected the fire records. Such a comparison of different fire markers in Greenland ice cores is a worthwhile goal, although the authors should highlight that some of their assumptions and conclusions are specific to Greenland. The first two-thirds of the paper are detailed and provide well-rounded explanations, while the final third feels rushed with a conclusion of the necessity of extracting more ice cores across Greenland for better spatial representation of these markers.

The current studies of biomass burning in ice cores are exponentially increasing. I applaud the authors for systematically investigating the differences between these proxies in a specific region as such comparisons help the ice core and fire science communities as a whole. This paper adds a beneficial review to the literature.

General comments:

The authors have extensive knowledge and a corresponding publication record of investigating ammonium in Greenland ice cores. The authors therefore strongly emphasize ammonium throughout this investigation of biomass burning proxies in Greenland ice cores and compare every other chemical fire marker to ammonium (e.g. Figure 2 and all sections of 3.2). However, the authors should justify why they use ammonium as the marker against which they compare every other record. The paper circuitously mentions (but not until section 3.3) that ammonium is mainly a viable fire marker in the poles as these regions are located far enough away from biogenic sources of ammonium, but this important point should be mentioned very early in the introduction. The authors could strengthen the entire paper if they are able to demonstrate that biogenic ammonium does not influence the ammonium record in Greenland. *OK, we add at the end of section 3.2 (i.e. before any comparison between ammonium and other species): "In the following sections, we compare the different potential biomass-burning proxies to ammonium. In addition to the fact that many more high-resolution records are available for ammonium than for any other chemical species, the choice of ammonium as the reference species is legitimate since its non-biomass-burning background summer level in Greenland ice is relatively low (less than 18 ppb; Legrand et al., 1992). Estimates of the non-biomass-burning ammonium level were derived from simultaneous comparisons of formate and ammonium levels, exploiting the fact that forest fire debris reaching Greenland mainly consists of ammonium formate and that formate background levels in Greenland ice exhibit low temporal variability. These features are confirmed in Fig. 2a and b. »*

Although the authors provide an overview on the known atmospheric lifetime of many of the species, the authors assume that each proxy will follow the same back trajectory. Some markers (ie vanillic acid) may be able to be injected higher into the atmosphere than heavier particles such as BC. *Well the parameter controlling the lifetime is not the weight of the species (all species like BC, and other biomass burning aerosol transported over large distances are submicron size particles).* These different injection heights can influence their transport and thereby also influence their atmospheric lifetime. The authors mention the difference in plume altitude with respect to the relative abundance of NO compared to NH3,

but (page 8) but do not mention this important parameter when investigating the contributions of the other biomass burning markers. A discussion on injection heights and depositional differences between markers should be included in the paper. *That is basically not possible since, except gaseous formic acid, most of organic markers (at the opposite to case of ammonia and NO) are not yet documented versus heights.*

Section 4.1.2 distracts the reader from the central aims of the paper as the level of detail included in this section is much greater than that provided in other sections, and as the evidence for or against the Tunguska asteroid impact is only obliquely related to biomass burning. Significantly shortening or even omitting this section will help "tighten" the manuscript. *We agree that the discussion on the Tunguska in the main text can distract the reader. On the other hand, the record of this event in ice cores has been very controversial since many years, specially within the ice core community. Here, because we simultaneously checked many chemical compounds (nitrate, ammonium, but also organics), we can, for the first time, discuss in detail the different possibilities. We therefore put the Tunguska discussion in an Appendix.*

Specific comments:

Page 2, Lines 4 and 5: What spatial and temporal differences occurred since 1900? *OK we reworded this sentence as "Whereas the mean burned forest area is expected to increase in the future in boreal regions (Flannigan et al., 2005), the change will not be spatially uniform and may differ from western to eastern North America and from North America to Siberia (Flannigan et al., 2001; Girardin et al., 2009)."*

Page 2, Lines 9 and 10: The first known human-origin fires occurred 1 million years ago (Berna et al., PNAS, 2012). Although he scientific community is actively seeking 1 million year old ice, currently the record of human-origin fires is longer than any ice core record. Please revise or omit this statement. *Yes but the Berna' study is for South Africa and in lines 9-10 we discuss high northern latitudes. To be clearer we reworded a bit as "Thus, we need to examine climate, fire conditions, and vegetation interactions through time prior to the appearance of human-origin fires at high northern latitudes to understand both natural and human impacts on past and present burning and improve predictions of future fire activity. »*

Page 2, Lines 20-23: Mentioning that paleo-fire records are available for these distinct regions of North America is somewhat misleading. Using the publicly-available R paleofire package (Blarquez et al., Computers & Geosciences, 2014), scientists are able to compile charcoal data into any regions of their choosing. *OK and we reworded this sentence as: "In this way, paleo-fire records have been obtained at regional, continental, and global scales and can be compiled from the paleo-fire database for specific regions (Blarquez et al., 2014). Published North American paleo-fire compilations covering the Holocene, for example, already are available for four distinct regions (Northwestern boreal, St. Lawrence, western U.S., and central North America) (Marlon et al., 2013). "*

Page 3, Lines 15-17: Do you mean that high-resolution records are limited form the present day until the last millennium OR that these high resolution records only exist for the past millennium? *OK we clarified this point: "Because ammonium (but not formate) can be measured with CFA, high-resolution records of ammonium and formate are few in number and limited to the last millennium since IC measurements and sub-annual sampling are required (Legrand and De Angelis, 1996; Savarino and Legrand, 1998)."*

Page 5, Lines 37-29: Please state why it is important that nitrogen emissions are dominated by ammonia and not NOx. *OK the explanation is given in the next sentence so we changed the text as "More recently, data have become available for some species distinguishing between temperate and boreal fires (Akagi et al., 2011). The case of nitrogen emissions is a good example of the need to get data distinguishing temperate from boreal fires. Indeed, whereas Andreae and Merlet (2001) reported an emission factor two-fold higher for NO than for $NH_3$ (3 g of NO against 1.4 g of $NH_3$ per kg of DM) for extratropical fires, Akagi et al. (2001) reported emission factors of 2.7 g for $NH_3$ against 0.9 g for NO per kg of DM for boreal fires (0.8 g of $NH_3$ against 2.5 g for NO per kg of DM for temperate fires). Thus, one of the most important emission features from boreal fires lies in nitrogen emissions dominated by ammonia but not $NO_x$ emissions."*

Page 6, Line 6-10: Do you mean that burning is not expected to produce levoglucosan in flaming conditions regardless of the vegetation type? *Yes, see Gao et al. (2003) as referenced in the text.*

Page 8, Line 1-6. If the R2 of formate versus ammonium is higher than the R2 of vanillic acid versus ammonium, why then use vanillic acid? Do you prefer vanillic acid because formic acid is produced during smoke plume aging (page 7)? Why do you not show formate in Figure 6? *In fact, as stated in section 3.2, formate data are only available at Summit. Figure 6 refers to D4 for which the only carboxylate data are for vanillic acid (not formic acid). We clarify this point in the revised version: "Although present at concentrations well below those of organic compounds (less than 1 ppb) discussed above, vanillic acid was investigated using CFA in Greenland snow layers (e.g, at D4; McConnell et al., 2007) with the aim of attributing the contributions of fossil fuel and biomass burning to the budget of BC during the two last centuries. We report annual levels of vanillic acid and ammonium in snow layers deposited at D4 between 1740 and 1870 (Fig. 6). Although formate was not measured in D4 snow, comparisons of temporal variability indicate that vanillic acid also can be used as a surrogate of ammonium ($R^2$ = 0.62 for vanillic acid versus ammonium, compared to $R^2$ = 0.87 for formate versus ammonium at Summit). "*

Section 3.2.5. The authors state studies that are towards the low end of the spectrum of atmospheric lifetimes for levoglucosan. Slade and Knopf, 2013 mention that under atmospheric background conditions, levoglucosan likely has an atmospheric lifetime of 2 weeks, while Bai et al., 2013 suggest that levoglucosan may have a mean atmospheric lifetime of 26 days. *Well, in fact Slade and Knopf (2013) report an atmospheric lifetime of 2 days to 2 weeks (not 2 weeks). The even larger atmospheric lifetime proposed by Bai et al. (2014) is based on quantum chemical calculations between 200-1500K in the gas phase (these calculations in discussing the lifetime of levoglucosan during the combustion process but not afterwards when levoglucosan is mainly in the gas phase).*

*Anyway thank you for this remark. We add these two references in the revised manuscript as follows: "These laboratory studies suggest an atmospheric lifetime of levoglucosan against chemical degradation of approximately two days (Lai et al., 2014), or from two days to two weeks (Slade and Knopf, 1013). Given the lower range of estimated levoglucosan lifetime against chemical degradation we cannot rule out that chemical loss represent a significant loss for levoglucosan additional to the depositional loss that would apply to all biomass burning aerosol."*

Section 3.2.6: Why do you designate these different groups if you do not use them later in the paper? Mentioning that certain markers are "grouped" or "similar" rather then designating specific groups (ie Group 1) may help the reader. *We don't really understand your comment: we denoted this group of species in view to avoid repeating in this paragraph "ammonium, formate, OC (DOC or TOC), BC, vanillic and glycolic acids".*

Section 3.3: Why do you not include the Akedemii Nauk (Siberia) ice core? This core may provide a link between the Greenland cores and the more temperate Siberian and Alaskan core that you include in the paper. *For two reasons. First, due to the relatively low altitude of the ice cap (724 asl), the AK Nauk ice core shows evidence of summer melt and infiltration processes (Opel et al., J. Glaciol., 55, 21-31, doi:10.3189/002214309788609029, 2009) which may seriously influence the atmospheric records preserved in the ice (Fritzsche et al., 2005). Surface melting occurs almost every year when temperatures may rise above 0°C even at the ice cap summit, and a considerable amount of the Akademii Nauk ice core consists of melt layers and partly infiltrated firn (Opel et al., 2009). Therefore, this ice core does not help for our discussion of sporadic summer events. Second, as discussed by Spolaor et al. (tc, 10, 245-256, 2016) 6-days backward trajectories indicate that in summer the quasi-totality of air masses have travelled over the Arctic ocean and very rare are back trajectories having been in contact with the Siberian forest.*

Page 13: If there is little difference between the 0, 250 and 500 m above ground level back trajectories, why do you choose the 500 m trajectories? You are investigating material that is deposited at the surface (0 m above ground level). *Yes, but on the other hand, we have also to consider that aerosol in trapped by clouds located a few hundreds m above the surface. Anyway the results are very similar.*

Figures 11 and 12: Although you mention the source of your data in the paper, please also mention this data source in the figure captions. *OK Done*

Page 16 Lines 34-39: The authors spent multiple pages in the beginning of the paper outlining differences between what individual proxies record (ie BC results from flaming fires and/or fossil fuel burning while levoglucosan is produced from smoldering fires and where ammonium contains a biogenic emissions source). I agree that "more work is needed to elucidate why levoglucosan (and neither BC nor ammonium) would be able to record Siberian fire activity in Greenland ice " is necessary, but the authors themselves provide plausible explanations earlier in the paper. In Section 4.3 the authors mention that levoglucosan may "mirror changes of fire activity at a larger scale. . .. than ammonium records" which is another explanation for these differences. The authors should include these possibilities during the discussion in Section 4.2 (lines 34-39). *Indeed it is what we would conclude from the derived different ice core trends (levo versus ammonium or BC). However, we cannot propose a reason for that since these different trends (a larger spatial scale for levo) tend to conflicts with a chemical degradation of levoglucosan. So we added: "Note that this conclusion conflicts with the hypothesis of a significant chemical degradation of levoglucosan suggested by laboratory studies (Sect. 3.2.5). More work is needed to elucidate why (1) levoglucosan might reflect Siberian fire activity in Greenland ice but not BC or ammonium, and (2) the finding that the levoglucosan record seems to mirror changes of fire activity at a larger spatial scales (Eurasia plus Canada) than ammonium records. «*

I completely understand the difficulties in writing an academic manuscript in another language. However, in many locations in the paper the writing style gets in the way of

understanding the science (e.g. page 5). Multiple co-authors are native English speakers and I strongly urge these co-authors to carefully edit the paper to remove instances of passive voice, flipped clauses and noun-adjective pair placements. *OK, a check-up of the whole manuscript has been done.*

Technical corrections:

Page 3, Line 3: Place "biomass burning source" or some equivalent after "this". *OK Done*

Page 3, Line 39: Place "comparison" after "this". *OK Done*

Sometimes temperatures are recorded in degrees C and sometimes in degrees K. Please be consistent throughout the paper. *OK Done, all temperatures are now in °C.*

Page 8, Line 17: Replace "unit" with "unity" *OK Done*